# Green synthesis of propylene oxide directly from propane

Pierre Kube[1], Jinhu Dong[1], Nuria Sánchez Bastardo[2], Holger Ruland [2], Robert Schlögl[1,2], Johannes T. Margraf [3], Karsten Reuter [3] & Annette Trunschke [1] ✉

The chemical industry faces the challenge of bringing emissions of climate-damaging $CO_2$ to zero. However, the synthesis of important intermediates, such as olefins or epoxides, is still associated with the release of large amounts of greenhouse gases. This is due to both a high energy input for many process steps and insufficient selectivity of the underlying catalyzed reactions. Surprisingly, we find that in the oxidation of propane at elevated temperature over apparently inert materials such as boron nitride and silicon dioxide not only propylene but also significant amounts of propylene oxide are formed, with unexpectedly small amounts of $CO_2$. Process simulations reveal that the combined synthesis of these two important chemical building blocks is technologically feasible. Our discovery leads the ways towards an environmentally friendly production of propylene oxide and propylene in one step. We demonstrate that complex catalyst development is not necessary for this reaction.

The discovery of new heterogeneous catalysts requires creative approaches to materials synthesis[1]. Since the optimal catalyst performance of the resulting solids often develops under very different process conditions in each case, the search for better catalysts requires the simultaneous variation of the reaction conditions over a wide range[2,3]. Depending on the temperature, the pressure or the composition of the feed gas, different active phases can form from the same freshly prepared catalyst precursor. Final and intermediate products can also react with and modify the catalyst. Both reversible dynamic and irreversible (surface) reconstructions, as well as the frequently very complex reaction networks entangled with catalyst properties, are responsible for the enormous intricacy that must be taken into account in the development of improved functional materials based on chemical intuition or descriptors and Big Data[4,5]. Herein, we show that this material optimization can be largely bypassed for the coupled synthesis of the valuable oxidation-sensitive products propylene and propylene oxide from propane by optimizing the process conditions rather than the sorption properties of the gas-solid interface. Instead of a redox-active

catalyst, inert fillers such as boron nitride, silicon oxide or silicon carbide are loaded into the reactor to improve mass and energy transport of the reaction that occurs in the gas phase.

Propylene, as a key building block of the chemical industry, was produced globally at a scale of 130 million metric tons in 2019[6]. Propylene oxide is a major intermediate for the synthesis of a large variety of consumer products including polyether polyols that are used in the manufacture of polyurethanes, propylene glycols as raw materials for the production of unsaturated polyester resins applied in the textile and construction industries, and propylene glycol ethers utilized as solvents in paints, inks, coatings, and many other related applications[7,8]. Large-scale production of propylene oxide (globally ~10 million t/y in 2012) starts from propylene produced from crude oil fractions by steam cracking or fluid catalytic cracking (FCC). About 10% of all produced propylene is used for the manufacture of propylene oxide using three main production technologies including the chlorhydrin process, coproduct routes, and liquid-phase epoxidation of propylene with hydrogen peroxide in methanol as solvent at 30–80 °C and 10–30 bar over a titanium silicate (TS-1) catalyst[7,9]. The

[1]Fritz-Haber-Institut der Max-Planck-Gesellschaft, Department of Inorganic Chemistry, Faradayweg 4-6, 14195 Berlin, Germany. [2]Max-Planck-Institut für Chemische Energiekonversion, Department of Heterogeneous Reactions, Stiftstrasse 34-36, 45470 Mülheim an der Ruhr, Germany. [3]Fritz-Haber-Institut der Max-Planck-Gesellschaft, Theory Department, Faradayweg 4-6, 14195 Berlin, Germany. ✉e-mail: trunschke@fhi-berlin.mpg.de

need of expensive auxiliary chemicals such as hydrogen peroxide, the complexity of the processes, and considerable environmental burdens due to waste formation imply economic drawbacks of the current multi-step production technologies. Hence, direct synthesis of propylene oxide using molecular oxygen attracted great interest[10], and the challenging task initiated ground-breaking research[11–13]. Supported Ag catalysts modified by promoters and $TiO_2$-based systems exhibit good prospects in heterogeneous oxidation of propylene by $O_2$. However, although the activity of the $TiO_2$-based catalysts is higher compared to Ag systems, the selectivity to propylene oxide is too low for an industrial application in both cases[10]. In summary, sensitive products such as propylene oxide are usually synthesized at highly specific catalyst surfaces where molecules interact strongly.

In this work, we show that the fast reaction at non-specific interfaces can be successfully used for the synthesis of products prone to overoxidation. We have performed kinetic studies on the oxidation of propane to propylene and propylene oxide on supposedly inert filler materials and have demonstrated by microkinetic modeling that the reaction proceeds in the gas phase. An estimative techno-economic assessment revealed that the direct production of propylene and propylene oxide from propane has indeed potential for technical application compared to conventional industrial processes of propylene oxide production.

## Results and discussion

### Selective oxidation of propane

In our investigation of transition metal-free catalysts in direct oxidation of alkanes, hexagonal boron nitride was reconsidered that has been reported to show activity and high selectivity in the oxidative dehydrogenation of propane to propylene[14,15], and in oxidative dehydrogenation of other substrates[16–18]. We compared the performance of $h$-BN in propane oxidation with other supposedly inert materials, like crystalline silica (sea sand, α-quartz), amorphous fumed silica (Aerosil 380), and silicon carbide (Supplementary Table 1). The specific surfaces and pore volumes of the materials vary over a wide range. All materials contain trace impurities of transition metal and main group elements, but the concentration is very low and differs from material to material.

The conversion of propane under the same reaction conditions depends on the material loaded (Fig. 1), but, unexpectedly, all materials show identical product selectivity. We also surprisingly discovered that a mixture of propylene and propylene oxide (two valuable products) formed above the apparently inert powders.

Interestingly, the formation of propylene oxide has never been reported in studies of propane oxidation with molecular oxygen over either redox-active selective oxidation catalysts or nitrides and carbides, although it is well known that PO and numerous other intermediates are part of the reaction network in the low-temperature (<630 °C) ignition chemistry of hydrocarbons in the homogeneous gas phase, which involve peroxy and hydroperoxy radical species[19–21]. In a very similar way, hydrogen peroxide was obtained in quite high yields in propane oxidation at 430–450 °C in empty steel reactors together with propylene, formaldehyde, acetaldehyde, methanol, and propylene oxide[22]. There is, however, a report on the formation of propylene oxide in propane oxidation at temperatures between 400 und 430 °C on $SiO_2$-supported $V_2O_5$ catalysts[23]. But in this case, $N_2O$ was used as oxidant, instead of molecular oxygen. The process conditions and the filling materials applied in the present study apparently allow the stabilization of propylene oxide, which is an instable intermediate in total oxidation of propane in the presence of oxygen on redox-active catalysts.

Propylene, propylene oxide, ethylene, and CO are the main products in the parameter space investigated, while $CO_2$, acetaldehyde, acrolein, and propionaldehyde are formed in negligible amounts (Fig. 1 and Supplementary Fig. 1). When propylene oxide forms, the oxygen

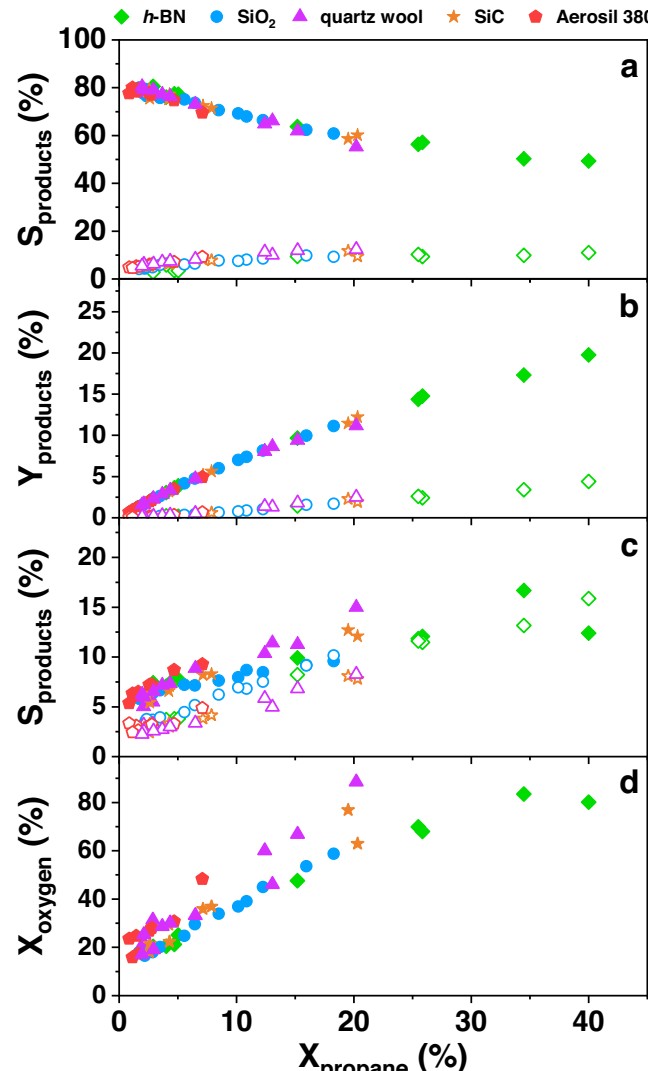

**Fig. 1 | Performance of propane oxidation using various filling materials in the reactor. a** Selectivity and **b** yield of propylene (filled symbols) and propylene oxide (open symbols); **c** selectivity to ethylene (filled symbols) and CO (open symbols), and **d** conversion of $O_2$ shown as a function of propane conversion; The data are measured in a tubular fixed bed reactor with the following reaction conditions: $T = 470$ °C–510 °C, feed ($C_3H_8/O_2/He = 30/15/55$), flow rate (3.3–20 ml/min), mass was 0.188 and 0.376 g for $h$-BN, 0.3, 0.6, and 1.0 g for $SiO_2$, 0.5 g for SiC, and 0.051 g for Aerosil 380.

concentration in the gas phase is not zero (Fig. 1d). The similarity in the product distribution over all apparently inert substances indicates that the reaction network is identical in all experiments. Differences between the different filling materials can only be observed with regard to the propane conversion, which could be attributed to the different thermal conductivity of the materials (Supplementary Table 1). A very similar observation was made by Hermans and co-workers in their investigation of boron-containing catalysts, such as boron carbide, titanium boride, nickel boride, cobalt boride, hafnium boride, and tungsten boride and elemental boron, however, without the formation of propylene oxide being reported[24]. In this work and numerous other studies, surface-stabilized $BO_x$ species have been thought to be the active sites in the oxidative dehydrogenation of propane at the solid-gas interface.

The apparent activation energies $E_a$ (Supplementary Table 2 and Supplementary Fig. 2) are much higher than those measured for vanadium oxide catalysts[25]. The latter were most frequently

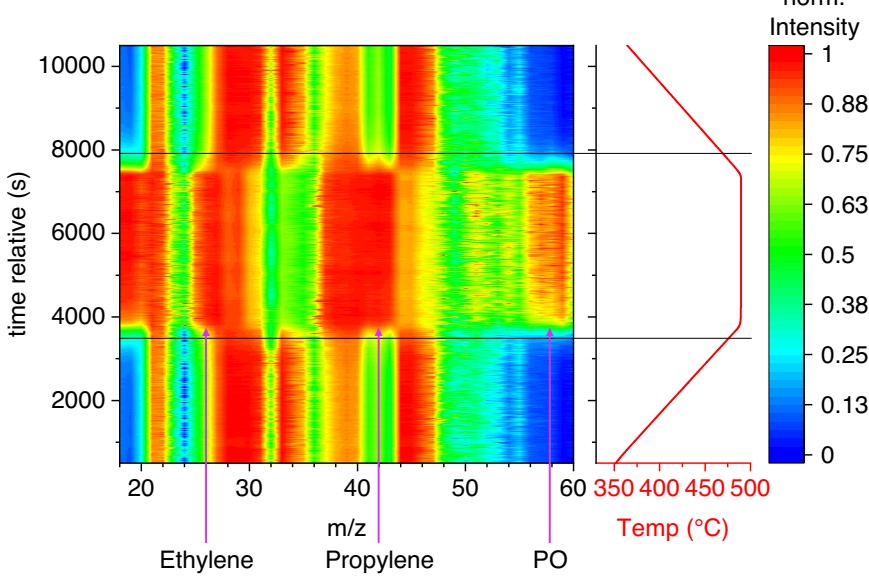

**Fig. 2 | Temperature-programmed oxidation of propane using SiO₂ as filler.** Reaction conditions: $T = 350\,°C–490\,°C$, total flow = 10 ml min⁻¹, $m = 670$ mg, feed ($C_3H_8$/$O_2$/He) = 30/15/55, heating rate 2.5 K min⁻¹; Mass-to-charge ratios m/z 18 to 60 were recorded.

determined for the consumption rate of propane and vary between 40 and 170 kJ/mol. The data from our study are consistent with the values measured previously in concentrated feeds over boron nitride[14,26]. In oxygen-rich feed, on the other hand, values around 190 kJ/mol were found[27–29]. The observations made by Loiland et al. in propane rich feed led to the conclusion that the catalysts seem to initiate the formation of radicals on their surfaces under non-dilute conditions, which then desorb into the gas phase to undergo radical chain reactions[26]. The contribution of radical reactions to the reaction mechanism via boron-containing catalysts has since been increasingly discussed[30–32].

In addition to the unusual high $E_a$, very high reaction orders with regard to propane of 2.3 for h-BN and 2.8 for SiO₂ were measured. This means that the rate of the reaction strongly depends on the concentration of propane in the gas phase. In contrast, values of <1 are frequently determined in propane oxidation over vanadium oxide-containing catalysts[33]. The partial pressure of oxygen has only a small influence on the rate (reaction orders 0.4 for h-BN and 0.3 for SiO₂), similar to redox-active catalysts[33]. The stability of the performance has been proven for SiO₂ for more than 100 h (Supplementary Fig. 3).

An increase in the propane concentration causes an increase in the integral formation rates of propylene oxide and ethylene (Supplementary Figs. 1 and 4). The formation of CO and CO₂ is hardly affected (Supplementary Fig. 1). The results show that a higher concentration of propane is needed to reach a higher PO selectivity.

The propane conversion increases with increasing layer height of the material, but there is no linear relationship (Supplementary Fig. 5). Therefore, a limitation of the reaction rate by film diffusion can be excluded. Furthermore, rate limitations due to pore diffusion are considered unlikely due to the non-microporous nature of the used materials. These results clearly indicate that the surface of the different materials is not directly catalyzing the reaction, but their involvement in initiating or quenching radical reactions cannot be excluded.

**Temperature-programmed experiments**

Our observation that SiO₂, SiC, Aerosil 380 and a reactor filled with quartz wool show very good performance in the propane oxidation (Fig. 1), similar to published boron-containing materials, although propylene oxide formation was also observed for the first time in the present work, suggests that gas-phase reactions are taking place instead of surface catalyzed reactions[19,20,31,34]. The kinetics of

elementary reactions in low-temperature autoignition chemistry of alkanes have been reviewed by Zador et al.[20]. Kinetic models for the ignition and combustion of propane in air and for the oxidation of propylene in the gas phase can be found in the literature, e.g., by Titova et al.[35] and Wilk et al.[34]. Basically, all the intermediates observed in the present work can be explained by reactions that are part of the extensive and complex reaction networks proposed in the combustion chemistry of propane in the gas phase[20,34–37]. To verify whether gas phase processes dominate in our case, temperature-programmed experiments were carried out in which the gas formed was analyzed by mass spectrometry (Fig. 2).

The reaction conditions chosen in the temperature-programmed experiment correspond in part to the conditions that have also been used in the literature when investigating boron-containing catalysts. The gas composition does not change over a wide temperature range (350 °C to approx. 470 °C). At a temperature of about 470 °C, however, propylene suddenly appears (an increase of m/z 42), while propane is consumed at the same time (decrease of m/z 44). After the reaction reaches its full extent, products such as water and propylene oxide are observed as molecular ion peaks (increase in m/z of 18 and 58, respectively). At the same time, an increased intensity is observed in the m/z range from 25 to 27, which could be attributed to the formation of ethylene or molecular fragments of the propylene formed. The consumption of oxygen can be clearly seen in the decreasing intensity of m/z 32.

The course of the reaction definitely speaks for an ignition of the propane-oxygen reaction mixture, which triggers the corresponding gas phase reactions. The ignition of alkane-oxygen mixtures on noble metal catalysts has been intensively studied mainly with regard to partial oxidation to synthesis gas, concluding that the surface mediates the ignition[38–41]. The ignition behavior differs only slightly when boron nitride is filled into the reactor instead of SiO₂ (Supplementary Fig. 6) and it can be controlled via the reaction conditions, such as the heating rate (Supplementary Fig. 7). Filling the reactor with an inert material apparently supports gas mixing and heat transport, but is not absolutely necessary for product formation as observed when using a completely empty reactor (Supplementary Fig. 8). It is important here that there is no redox-active catalyst in the reactor, which can lead to subsequent reactions of less stable products, such as propylene oxide, with the oxygen that is still present (Fig. 1d) and thus to the total

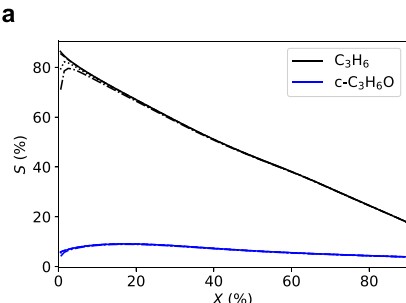

**a**

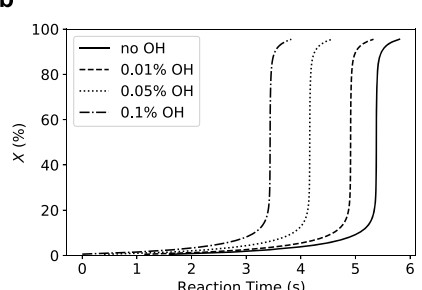

**b**

**Fig. 3 | Microkinetic simulation of the gas-phase reaction. a** Selectivity (S) with respect to the formation of propylene ($C_3H_6$) and propylene oxide (c-$C_3H_6O$), versus conversion of propane (X); Shown are results for microkinetic simulations of an ideal gas reactor at 500 °C, using a gas-phase mechanism[37] and a feed composition of 30% propane and 15% oxygen; **b** Conversion of propane (X) as a function of reaction time, for the simulations in **a**; The different traces correspond to the fraction of OH radicals introduced to the feed (solid: 0.0%, dashed: 0.01%, dotted: 0.05%, dash-dotted: 0.1%); The remaining fraction of the feed is composed of $N_2$.

oxidation of valuable products to $CO_2$. The gas-phase oxygen itself is essential for the initiation and propagation of the radical chain reactions that lead to valuable products.

## Microkinetic simulation

To obtain deeper insights into the mechanism of propylene oxide formation in these experiments, we turned to microkinetic models of this process. As already noted by Kraus and Lindstedt[31], the experimentally observed selectivity towards propylene is fully consistent with a gas-phase mechanism for the conversion. Meanwhile, the predicted selectivity for propylene oxide is somewhat sensitive to the details of the underlying microkinetic model. With the experiments presented herein, this uncertainty can be clarified, as we find good agreement between experimentally observed selectivity and those predicted for an ideal gas reaction when using the recent "DTU" mechanism for propane oxidation[37]. Importantly, this mechanism (unlike the others considered in ref. [31]) includes hydroperoxo alkyl chemistry that becomes relevant in the temperature range of this process[21].

The good agreement (Fig. 3a) of these ideal gas simulations with the experiments and the fact that experimental selectivity is basically independent of the employed material in the reactor thus clearly point to a gas-phase mechanism of the conversion. This raises the question why different materials nonetheless display different activities. A possible explanation for this is that the materials modify the composition of the gas mixture, i.e., generate radicals that accelerate the gas phase chemistry or quench radicals[42,43]. To check this hypothesis, we ran additional simulations with modified feed compositions, containing small fractions of •OH (Fig. 3b). Here, •OH is an exemplary radical species that could plausibly be formed from the couple product water on $h$-BN or $SiO_2$ surfaces[44–46]. For example, it was found that an increased concentration of OH species on the surface of BN leads to a very productive catalyst[47]. We find that this modification only slightly affects the selectivity at low conversions (Fig. 3a). However, as seen in Fig. 3b, propane conversion is significantly accelerated by the presence of these radicals.

These simulations thus point to a pure gas-phase mechanism for the conversion of propane to propylene and propylene oxide. The filling materials can nonetheless play an important role, by generating radical precursors that can significantly accelerate the conversion. Activities could potentially be further increased by tuning the feed composition and the filling material and by optimizing the process technology.

## Economic efficiency assessment

In summary, unexpectedly high yields of propylene (20%) and propylene oxide (4.4%) can be achieved in a propane rich feed (30% propane or higher and 15% oxygen) (Fig. 1) at a reaction temperature of 490 °C.

The economic viability of a hypothetical one-step process was estimated using Aspen HYSYS (Supplementary Fig. 9). Figure 4 shows the flowchart of the direct process (40 % propane conversion at 490 °C and 1 bar, 11 % propylene oxide selectivity), which is the basis for the Aspen simulation model, in comparison with three common current processes for manufacturing of propylene oxide under the assumption that these also start from propane in order to enable a holistic view.

Considering the benefit of the other valuable products produced directly from propane in such a direct process according to their market prices (propylene 1048$/ton[48], hydrogen 6000$/ton[49], and acetaldehyde 973$/ton[50]) propylene oxide could be sold for the hypothetical price of 3570$/ton, which is close to the current market price of 2807$/ton[51]. A more detailed description of the process simulation is given in the Supplementary Information. It should be mentioned here that the cost/economic estimation can be optimized to represent industrial conditions. Possible parameters for optimization are the utilization of process heat, the use of air instead of a helium-oxygen mixture as feed gas, which was used under our laboratory conditions, and the optimization of the distillation columns.

The direct process from propane to propylene and propylene oxide therefore definitely has potential for industrial application. The process would be absolutely green if the energy supply is provided by electrical renewable energy, because hardly any climate-damaging $CO_2$ is produced as a by-product. In our experiments, the ignition of the propane-rich reaction mixture at about 470–490 °C was crucial for the formation of products that can easily be further oxidized. This teaches us that we can produce thermodynamically less stable, oxidation-sensitive products in oxidation reactions if we set conditions for fast reactions and omit the catalyst.

The knowledge gained is of far-reaching importance for research in oxidation catalysis, as many catalysts including vanadium oxide-based materials are investigated in this temperature range[25]. Reaction mechanisms assuming surface reactions, as proposed for boron oxides, can be ruled out[14,52]. Redox-active catalysts that give rise to activated oxygen species on their surface are against a long tradition of searching for selective complex interfaces rather inappropriate for the formation of sensitive reaction products, such as propylene oxide[25]. We think that it is easier to generate propylene oxide in propane-rich feed composition and at a temperature of 490 °C in the gas phase than to optimize a redox-active catalyst and the process conditions so that subsequent reactions of propylene oxide on the surface does not occur. The thermodynamically limited activation of di-oxygen to only superoxide and the fast sorption-desorption kinetics are success factors. Efforts to control the oxidative potential of atomic oxygen species, which can only form at surface defects, are less successful, possibly because the required high selectivity demands high sorption specificity and thus long residence times on the surface, leading to further combustion of the valuable reaction products.

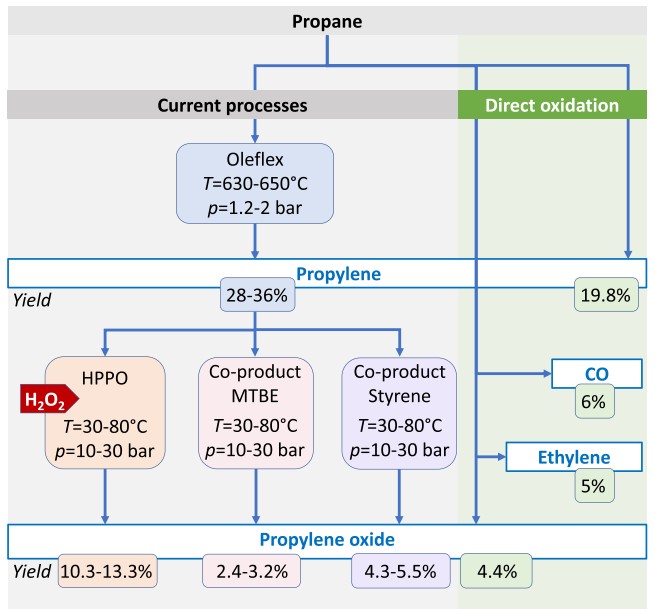

**Fig. 4 | Comparison of the yields of a direct propane oxidation route compared to established processes.** The calculation of the yield of valuable products was based on the same C3 source (propane). The established processes are based on propylene, which is previously produced by cracking petroleum fractions. Expensive additives and reactants such as $H_2O_2$ or solvents are used. For a direct comparison, it was assumed that propane, as found in natural gas, is also the feedstock for the established processes[7]. The calculation of the yield in single-pass illustrates the sustainability of the direct oxidation route with regard to the utilization of the carbon source. However, the decisive factor for the economic viability of the direct oxidation process is the sales price of PO simulated for it (see text).

## Methods
### Materials and gases
All used materials are commercially available and were ordered from the following companies: $SiO_2$ (Quartz) from Supelco (puriss p.a., LOT number SZBA0210, internal ID S28035), hexagonal boron nitride (h-BN) from Alfa Aesar (Quality 99.5%, LOT number E31M55, internal ID S25618), Aerosil 380 from Evonik (LOT number 157012015, internal ID S28106), and silicon carbide (SiC) from ESK-SiC GmbH (LOT number 654508, internal ID S32814). The gases propane (purity 99.95%), oxygen (purity 99.999%), helium (purity 99.999%) and nitrogen (purity 99.999%) were supplied by Westfalen company.

### Characterization of filling materials
Nitrogen adsorption was performed at −196 °C using the Autosorb-6B analyser (Quantachrome) after outgassing the catalysts in vacuum ($SiO_2$ and h-BN for 2 h at 200 °C, Aerosil 380 for 12 h at 200 °C, SiC for 2 h at 300 °C). All data treatments were performed using the Quantachrome Autosorb software package. The specific surface area $S_{BET}$ was calculated according to the multipoint Brunauer-Emmett-Teller method (BET) in the $p/p_0 = 0.05–0.15$ pressure range assuming the $N_2$ cross sectional area of 16.2 Å$^2$.

X-ray fluorescence spectroscopy (XRF) was used for elemental analysis applying a Bruker S4 Pioneer X-ray spectrometer. For sample preparation, the mixture of 0.1 g of the material and 8.9 g of lithium tetraborate (>99.995 %, Aldrich) was fused into a disk using an automated fusion machine (Vulcan 2 MA, Fluxana).

Inductively coupled plasma optical emission spectrometry (ICP-OES) was used as a second technique for elemental analysis. An Optima 8300 from Perkin Elmer with Zyklon nebulizer was used in axial mode. Minimum two points calibration with forced intercept at zero were measured with certified standards. Peak evaluation is based on three points per peak. Water was used as spectral blank. Dissolution of the samples was done in a multi-wave Pro autoclave from Anton Paar, equipped with Teflon liner at 200 °C and 60 bar. Reagents in supra pure quality and water from an ELGA pure water system (VEOLIA) (conductivity 0.05 μS/cm) were used.

Phase analysis was performed by X-ray diffraction (XRD) using a Bruker D8 ADVANCE diffractometer (Cu Kα radiation, secondary graphite monochromator, scintillation counter).

### Propane oxidation
The propane oxidation experiments were carried out in a self-built reactor with plug flow characteristics. All measurements were performed at atmospheric pressure and the pressure in the reactor was monitored with pressure sensors upstream and downstream of the reactor tube. The following general reaction conditions were applied: Mass of material from 100 to 1500 mg, temperature from 470 to 510 °C, total flow from 6.7 to 25 ml min$^{-1}$, propane from 10 to 60 vol%, oxygen from 1 to 15 vol%. Helium was used as balance. A certain amount of material (sieve fraction from 250 to 355 μm) was filled into the quartz reactor (inner diameter 7 mm) without dilution or the empty reactor was used. The gas hourly space velocity (GHSV), [h$^{-1}$] was calculated using the bulk volume of the material in the reactor $V_{material}$ and the applied volumetric gas flow at standard conditions ($T = 273.15$ K and $p = 0.1013$ MPa) $\dot{V}$ according to Eq. (1):

$$GHSV = \frac{\dot{V}}{V_{material}} \qquad (1)$$

The product gas mixtures were analyzed by online gas chromatography (Agilent 7890 GC). The following GC column combinations were used for product analysis: (1) Plot-Q (length 30 m, 0.53 mm internal diameter, 40 μm film thickness) plus Plot-MoleSieve 5 A (30 m length, 0.53 mm internal diameter, 50 μm film thickness), connected to a thermal conductivity detector (TCD) for analysis of the permanent gases (CO, $CO_2$, and $O_2$) and (2) Plot-Q (length 30 m, 0.53 mm internal diameter, 40 μm film thickness) plus FFAP (length 30 m, 0.53 mm internal diameter, 1 μm film thickness) connected to a flame ionization detector (FID) for analysis of hydrocarbons and oxygenates.

The calculation of the propane conversion ($X_{propane}$) and selectivity ($S_i$) of product i in percentage, were done based on the carbon number and the sum of all products using Eqs. (2) and (3), respectively:

$$X_{C_3H_8} = \frac{\sum_{i=1}^{n} N_i c_i}{\sum_{i=1}^{n} N_i c_i + 3c_{C_3H_8,out}} \times 100 \qquad (2)$$

$$S_i = \frac{N_i c_i}{\sum_{i=1}^{n} N_i c_i} \times 100 \qquad (3)$$

$N_i$ is the number of carbon atoms in product i, $c_i$ is the concentration of product i at the reactor outlet, and $c_{C_3H_8,out}$ is the propane concentration in the outlet gas.

The oxygen conversion was calculated using Eq. (4), where $c_{O_2,in}$ and $c_{O_2,out}$ are the concentrations of the oxygen in the feed gas at inlet and outlet position, respectively, of the reactor.

$$X_{O_2} = \frac{c_{O_2,in} - c_{O_2,out}}{c_{O_2,in}} \times 100 \qquad (4)$$

The yield ($Y_i$) of product i in percentage was calculated by using Eq. (5):

$$Y_i = \frac{X_{C_3H_8} \times S_i}{100} \qquad (5)$$

The carbon balance ($C_{balance}$) was determined according to Eq. (6):

$$C_{\text{balance}} = \frac{\sum_{i=1}^{n} N_i c_i + 3c_{C_3H_8,out}}{3c_{C_3H_8,in}} \times 100 \qquad (6)$$

In all experiments, the carbon balance was 100% +/− 5%. The formation of polymerization products was not observed.

Reaction rates $r_i$ for propane consumption, propylene formation and propylene oxide formation in mol $g^{-1}$ $h^{-1}$ were calculated using Eq. (7):

$$r_i = \frac{dn_i}{d\left(\frac{W}{F}\right)} \qquad (7)$$

The amount of starting compound i consumed or product i formed ($n_i$) was used in the unit mol $ml^{-1}$. $W$ is the mass of the material in g and $F$ is the total flow rate in ml $min^{-1}$.

Mass transfer limitations were excluded by measuring the propane conversion when using different amounts of material and different gas flows (see Supplementary Fig. 5) and checked by calculating the dimensionless Mears and Weisz-Prater criteria. $SiO_2$ has the highest Mears modulus of $5.7 \times 10^{-6}$ (must be $<1.8 \times 10^{-2}$) and Weisz-Prater modulus of $2.24 \times 10^{-3}$ (must be $<0.07$) for measurements at 490 °C of all materials tested, indicating that mass transport limitations do not play a role.

## Temperature-programmed experiments

The experiments were performed in the same reactor setup, which was described in the previous section. An online mass spectrometer (QMA 400, Pfeiffer Vacuum) was used for recording the reactant and product gas streams. 670 mg of $SiO_2$ and 665 mg of $h$-BN, respectively, were loaded into the reactor. A total flow of 10 ml/min, which was composed of 30% propane, 15% oxygen and 55% helium, was used. First, the reactor was heated up to 350 °C with a heating rate of 5 K $min^{-1}$ and hold at this temperature for minimum 15 min. Then the temperature-programmed experiment was performed by heating up to 490 °C with a heating rate of 2.5 K $min^{-1}$ or 5 K $min^{-1}$, respectively, holding at 490 °C for 1 h and then cooling down with the same rate like for heating up.

The reaction gas was withdrawn ~5 cm behind the material bed by using a capillary-vacuum pump combination and fed into the mass spectrometer (QMA 400, Pfeiffer Vacuum). All $m/z$ from 18 to 60 were monitored simultaneously with a scanning rate of 50 ms per $m/z$.

## Microkinetic simulation

The microkinetic simulations of gas-phase conversion were implemented with the Cantera package[53], using the "DTU" model of propane oxidation[37]. The technical settings of these simulations were analogous to those used by Kraus and Lindstedt[31]. Specifically, a constant pressure, ideal gas reactor was used and the time evolution of the gas mixture was modeled using a dynamic time step, adjusted by the solver. All simulations were performed at atmospheric pressure and a temperature of 500 °C.

## Process simulation

The direct oxidation of propane has been simulated using Aspen HYSYS. This process involves the reaction of propane with oxygen to produce propylene oxide as the target product. The rest of the considered reaction products include a mixture of propylene, ethylene, acetaldehyde, hydrogen, water, carbon monoxide and carbon dioxide. The Cubic-Plus-Association (CPA) package has been chosen as the fluid package for the simulation. The CPA property package uses the Cubic-Plus Association equation of state model and is suitable for the simulation of mixtures containing hydrocarbons, non-hydrocarbons such as carbon dioxide, nitrogen, and polar/associating chemicals such as

water, alcohols, glycols, esters or organic acids. The process flow diagram (PFD) corresponding to the direct oxidation of propane simulated in Aspen HYSYS is given in Supplementary Fig. 9. Three main parts can be distinguished, i.e., conditioning of the feed gases, reaction, and separation of the reaction products and recycling of the unreacted reactants.

The feed gases (propane, oxygen, and helium) are heated up to the reaction temperature (490 °C) by heat exchangers (E-100, E-101, E-102) placed in each of the reactor inlet gas lines. Thereafter, the make-up gas enters the reactor CRV-100, where the reaction between propane and oxygen takes place at 490 °C and 1 bar. Under these conditions the experimental conversion of propane is 40%. The reactor outlet stream consists of unreacted propane and oxygen, helium, propylene oxide, propylene, ethylene, acetaldehyde, hydrogen, water, carbon monoxide and carbon dioxide. Then, different separation stages are carried out to separate the different products and unreacted gases. After cooling the reaction products (E-103), a vapor, liquid and aqueous streams are split in the separator V-100. The vapor stream is mainly composed of ethylene, hydrogen, oxygen, helium, carbon monoxide, and carbon dioxide, whereas most propane, propylene oxide, acetaldehyde, and propylene are recovered in the liquid stream. The vapor stream is further cooled (E-104) and separated into two streams in the separator V-101. The resulting streams are a vapor mixture consisting of helium and hydrogen, and a liquid formed by carbon monoxide, oxygen, and minor amounts of other compounds, such as ethylene, carbon dioxide, propane, and propylene. Helium and hydrogen are separated in V-102 after cooling (E-105). The liquid stream obtained in V-101 is heated (E-106) to recover carbon monoxide and oxygen in the top stream of the separator V-103. After that, oxygen is separated from carbon monoxide in the distillation column T-104 and fed into the reactor together with a fresh oxygen stream. The liquid stream obtained in V-100 is subjected to successive distillation stages to recover propylene, acetaldehyde, propylene oxide and unreacted propane in individual streams. In the distillation column T-100, the light components that could not be separated in V-100 (mainly ethylene, and carbon dioxide) are recovered in the top stream. In the bottom stream, propane, propylene, acetaldehyde, and propylene oxide are obtained. The bottom stream from T-100 enters the distillation column T-101, where propylene is separated in the top stream. The bottom stream from T-101 is fed to the distillation column T-102. Here, the unreacted propane is recovered in the top stream and recirculated to the reactor. Finally, the bottom stream of T-102 is separated into acetaldehyde and propylene oxide in the distillation column T-103.

Supplementary Table 3 shows the recovery of the main compounds in their corresponding streams as well as their mole fractions. More than ca. 95% of propane, propylene oxide, hydrogen, oxygen, propylene, and helium, respectively, are separated and recovered in individual streams. They are high purity streams in which the mole fraction of the corresponding compound is ≥0.99, except for the case of the acetaldehyde stream, which has an acetaldehyde mole fraction of ca. 0.95.

The plant cost estimation for the direct oxidation of propane to produce propylene oxide based on the Aspen HYSYS simulation has been performed using the Aspen Process Economic Analyzer (APEA) integrated in Aspen HYSYS. The plant costs can be classified in two major categories, i.e., capital, and operating costs. The capital cost includes the equipment and installed costs. Both costs represent the major fraction of the total capital cost. On the other hand, the costs associated with raw materials and utilities like separation account for the main part of the operating costs. The capital and operating costs have been used to calculate the minimum price at which propylene oxide could be sold, assuming that the capital cost will be completely paid during the first 5 years of the operation of the plant assuming 8000 h of operation per year (Supplementary Table 4).

## Data availability

The data generated in this study have been deposited in the internal FHI AC/CATLAB Archive https://ac.archive.fhi.mpg.de/P51805; Public access is enabled. Source data are provided with this paper.

## Code availability

The code for the microkinetic simulation has been deposited in the internal FHI AC/CATLAB archive https://ac.archive.fhi.mpg.de/P51805; Public access is enabled.

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

## Acknowledgements

The authors thank Maike Hashagen for measuring the specific surface areas of the investigated materials, Dr. Frank Girgsdies for X-ray diffraction analysis, Dr. Olaf Timpe for chemical analysis, and Dr. Oluwatoyin Omojola for transferring data for the process simulation to the Max Planck Institute for Chemical Energy Conversion.

## Author contributions

P.K. performed the propane oxidation and the temperature-programmed experiments and analyzed the data. He was also involved in planning the experiments. J.D. performed a part of the experiments, contributed to the discussion, and provided support for data analysis. N.S.B. and H.R. carried out the economic feasibility study. H.R. wrote the methods section on process simulation. R.S. contributed to the discussion, data interpretation, concepts of experiments, and writing. J.T.M. performed, interpreted and described the microkinetic simulation. K.R. contributed to the discussion, data interpretation, and writing. The work was conceptually supervised by A.T., who also has written the main part of the manuscript.

## Funding

## Competing interests

The authors declare no competing interests.
