## [Peer Review File · Nature Communications]

REVIEWER COMMENTS

Reviewer #1 (Remarks to the Author):

Although BN and supported BN are comprehensively studied for propane dehydrogenation in recent years, the formation of propylene oxide was predicted but experimentally not reported before (to the best of my knowledge). Hence, this MS is indeed an extraordinary development in this field, which is reporting propane to propylene oxide for the first time.

They observed unexpectedly high yields of propylene (20%) and propylene oxide (4.4%) at a reaction temperature of 490°C. Their calculation of the yield indicates the sustainability of the direct oxidation route with respect to the utilization of the carbon source.

Some of my comments are,

1. "gas-phase reactions are taking place instead of surface catalysed reactions" need experimental proof. Why can propylene oxide not form via surface catalyzed reaction? This needs a detailed discussion.
2. Catalysis studies are not complete; the effect of gas flow, gas ratios, temperature, pressures on product yield and selectivity are not studied.
3. Details catalysis stability study for a long duration (say 100 hrs) is needed.
4. Mechanistic aspects of the catalytic process are weak (only TPD study). DRIFTS or SSNMR or some other spectroscopic studies will provide the required insights.
5. Statement "surface-stabilized BO_x species have been thought to be the active sites in a catalyzed reaction at the solid-gas interphase". This is recently studied in detail here: ACS Sus. Chem. Eng. 2020, 8, 16124.

This MS will have a direct impact on how PO is produced in the industry in the future. However, authors should carry out more detailed studies to get an in-depth insight into this process. Overall, this is an excellent MS and should be accepted for publication after the above-detailed studies.

Reviewer #2 (Remarks to the Author):

The manuscript by Kube et al. reports on the co-production of propylene oxide (PO) and propylene during the thermal oxidation of propane.

Although conceptually this sounds great, in reality, co-production of chemicals can be challenging as the growth rate of both rarely line up. Having said that, the observed chemistry is interesting as – to the best of my knowledge – nobody reported PO formation during these reaction conditions.

Comments:

- In the abstract and later in the manuscript the authors refer to BN and other materials like

SiO₂ as being inert. This is not in line with several reports showing that the surface of BN oxidizes under reaction conditions. What the authors probably want to say is that the surface does not play a role in the product evolution (because several materials result in similar product distribution).

- This fact is however at odd with results in *Angew. Chem. Int. Ed.* 2020, 59(38), 16527-16535. In this paper it is shown that the C₂/C₃ ratio is predominantly controlled by the H-atom that is abstracted in the propane substrate: iso-propyl radicals exclusively form propylene whereas n-propyl can also form ethylene upon the elimination of a methyl radical. The ratio of iso-propyl/n-propyl radicals that one can expect based on radical gas phase chemistry is 1.5/1 whereas the experiments suggest a value closer to 1, based on literature values for the gas phase radical chemistry (also in line with the secondary C-H bonds in propane being weaker than the primary C-H bonds). This suggests that there are additional pathways that lead to this isopropyl-to-n-propyl ratio, possibly on the surface. I do however understand the hypothesis of the authors that the surface would not play a role as several materials gave similar selectivity. I would encourage the authors to reformulate this statement.

- Page 5 line 101: the differences in propane consumption rates for the different materials is suggested to be caused by the differences in heat transfer properties of the materials. I do not find the data to judge this statement but it seems unlikely to me that heat transfer would play a role at these small scales and low conversions (something that is also in line with the absence of mass-transfer limitations, as verified by the authors).

- Page 6, line 126: similar observations were reported in *Org. Process Res. Dev.*, 2018, 22 (12), 1644-1652: the observed rate scales linearly with the overall bed volume (more precisely: total reactor contact time), but not with the BN-based contact time. That study also shows how the reaction rate initially increases linear with #BN (keeping the reactor volume constant) and then decreases (volcano-like behavior), indicating radical inhibition (termination) at higher #BN loadings. These observations seem to disprove that the material would not play any role.

- the process simulation is not described in a lot of detail and is not succinct. For instance, page 11, line 199: ...products could be sold for the following hypothetical prices... Does this mean that those numbers were just assumed, or calculated based on a process model (which at this stage would have significant error margins), and if so: what value margin was assumed?

The strong point of this manuscript is that the authors observed the formation of PO under these reaction conditions. The authors do however not elaborate on its formation mechanism. I also feel that it is a bit of a stretch to suggest that this approach could replace PO production via traditional pathways. From that perspective I do not think the manuscript is a good fit for *Nature Communications*.

Reviewer #3 (Remarks to the Author):

Green synthesis of propylene oxide directly from propane
2-17 refers to page 2, line 17. Etc

I. General comments

This is an interesting report that would be a valuable addition to the literature. There are a few comments and requests for clarification below.

II. Main comments / questions

- a. Fig 1: there appears to be some variation in selectivity performance, specifically in c & d. Is this due to the large spread in temperatures that were used (470 to 510 oC)? Perhaps the data could be further clarified by adding figures in the SI
- b. Fig 1: why is BN the only material studied at higher conversions? Is it possible to get the

same results by increasing residence time over the other materials?

c. Were blank tests (with only quartz tubes, no silica wool) run?

d. 98: PO forms only when there is incomplete consumption of O₂? Have experiments been performed on higher initial O₂ concentrations and selectivity evaluated?

e. 105: Are the BO_x active sites why higher conversions are reported for hBN only?

f. 111: Are similar activation energies expected for the different stable intermediates? (in relation to 211) or were they higher (117)?

g. 214: The sentence starting with "redox active catalysts" is difficult to interpret. It would seem that both approaches are rather inappropriate for the formation of PO?

h. 218: how can you prove that a selective catalytic surface will not give rise to additional levels of success? It is well known that controlling the residence time distribution in partial oxidation catalysis (to remove products quickly from the reactor without overoxidation) gives rise to better yields. It is possible to combine the concepts of selective surface reaction (high specificity for target molecule, low specificity for product molecule) along with short residence times.

i. Have other oxidative PDH researchers (for example, those studying BN) reported higher yields of propylene than reported here? If not, that would give further credit to the arguments presented here. Many of these prior research groups were likely making PO as a product but did not have the proper GC systems to detect & analyze?

j. Regarding the techno economic analysis, I do not think that the authors accounted for any separation costs. How could one separate the product mixture that contains O₂ in addition to hydrocarbons?

III. Stylistic comments or typos

a. 117: small letter "addition"

Reply to Reviewers' comments

We are very grateful for the time the reviewers invested in studying our work and for the many valuable comments. On this basis, we have further improved the manuscript. Changes made in the manuscript are highlighted in yellow in the revised version. Below are our detailed answers to all the reviewers' questions and comments. The reviewers' comments are reproduced in black font, our answers in blue font.

Reviewer #1 (Remarks to the Author):

Although BN and supported BN are comprehensively studied for propane dehydrogenation in recent years, the formation of propylene oxide was predicted but experimentally not reported before (to the best of my knowledge). Hence, this MS is indeed an extraordinary development in this field, which is reporting propane to propylene oxide for the first time.

They observed unexpectedly high yields of propylene (20%) and propylene oxide (4.4%) at a reaction temperature of 490°C. Their calculation of the yield indicates the sustainability of the direct oxidation route with respect to the utilization of the carbon source.

Some of my comments are,

1. "gas-phase reactions are taking place instead of surface catalysed reactions" need experimental proof. Why can propylene oxide not form via surface catalyzed reaction? This needs a detailed discussion.

Reply: Thank you, that is a very valid point. The similarity of the change in selectivity as a function of conversion (Fig. 1) for completely different materials is one reason that led to this conclusion. Furthermore, we observed the ignition behavior and the associated sudden formation of propylene oxide (Fig. 2). Ignition also takes place in a completely empty reactor (Supplementary Fig. 6 in the original Supplementary Information, Supplementary Fig. 8 in the revised version). The delayed ignition at 490°C in this case may indicate that the filled materials also have an influence on the heat and mass transfer or by initiating or quenching radical reactions.

To substantiate gas-phase chemistry, we have simulated the experimental data using a microkinetic model assuming that the reaction takes place exclusively in the gas phase. The course of the selectivity is perfectly reproduced by this pure gas phase model (new Fig.3 in the revised manuscript).

However, the objection of Reviewer 1 is highly justified, since we experimentally observe different activities for the different materials that we fill into the reactor. Thus, the nature of the filling material has an influence on the propane conversion. This could be caused, for example, by different properties of the various surfaces in quenching of gas phase reactions (M. Suh, P. S. Bagus, S. Pak, M. P. Rosynek, J. H. Lunsford, *The Journal of Physical Chemistry B* 2000, 104, 2736-2742.). Another reason could be the surface-mediated supply of radicals that accelerate the gas phase reaction (K. B. Hewett, L. C. Anderson, M. P. Rosynek, J. H. Lunsford, *Journal of the American Chemical Society* 1996, 118, 6992-6997.). Hence, different surfaces can generate for example OH radicals differently and react differently with OH radicals. Indeed, we were able to simulate a positive influence of OH radicals on propane conversion, without the selectivity being significantly affected by the

addition of small amounts of these radicals. These new results have been included on Pages 10-11 of the revised manuscript:

Microkinetic simulation

To obtain deeper insights into the mechanism of propylene oxide formation in these experiments, we turned to microkinetic models of this process. As already noted by Kraus and Lindstedt,³¹ the experimentally observed selectivity towards propylene is fully consistent with a gas-phase mechanism for the conversion. Meanwhile, the predicted selectivity for propylene oxide is somewhat sensitive to the details of the underlying microkinetic model. With the new experiments presented herein, this uncertainty can be clarified, as we find good agreement between experimentally observed selectivity and those predicted for an ideal gas reaction when using the recent “DTU” mechanism for propane oxidation.³⁷ Importantly, this mechanism (unlike the others considered in ref. 31) includes hydroperoxoalkyl chemistry that becomes relevant in the temperature range of this process.²¹

Fig. 3: Microkinetic simulation of the gas-phase reaction.

a, Selectivity (S) with respect to the formation of propylene (C_3H_6) and propylene oxide ($c-C_3H_6O$), versus conversion of propane (X); Shown are results for microkinetic simulations of an ideal gas reactor at $500^\circ C$, using a gas-phase mechanism³⁷ and a feed composition of 30% propane and 15% oxygen; **b**, Conversion of propane (X) as a function of reaction time, for the simulations in **a**; The different traces correspond to the fraction of OH radicals introduced to the feed (solid: 0.0%, dashed: 0.01%, dotted: 0.05%, dash-dotted: 0.1%); The remaining fraction of the feed is composed of N_2 .

The good agreement (Fig. 3a) of these ideal gas simulations with the experiments and the fact that experimental selectivity is basically independent of the employed material in the reactor thus clearly point to a gas-phase mechanism of the conversion. This raises the question why different materials nonetheless display different activities. A possible explanation for this is that the materials modify the composition of the gas mixture, *i.e.*, generate radicals that accelerate the gas phase chemistry or quench radicals.^{42,43} To check this hypothesis, we ran additional simulations with modified feed compositions, containing small fractions of $\cdot OH$ (Fig. 3b). Here, $\cdot OH$ is an exemplary radical species that could plausibly be formed from the couple product water on h -BN or SiO_2 surfaces.⁴⁴⁻⁴⁶ For example, it was found that an increased concentration of OH species on the surface of BN leads to a very productive catalyst.⁴⁷ We find that this modification only slightly affects the selectivity at low conversions (Fig. 3a). However, as seen in Fig. 3b, propane conversion is significantly accelerated by the presence of these radicals.

These simulations thus point to a pure gas-phase mechanism for the conversion of propane to propylene and propylene oxide. The filling materials can nonetheless play an important role, by generating radical precursors that can significantly accelerate the conversion. Activities could potentially be further increased by tuning the feed composition and the filling material and by optimizing the process technology.

- 37 Hashemi, H., Christensen, J. M., Harding, L. B., Klippenstein, S. J. & Glarborg, P. High-pressure oxidation of propane. *Proceedings of the Combustion Institute* **37**, 461-468, doi:<https://doi.org/10.1016/j.proci.2018.07.009> (2019).
- 42 Suh, M., Bagus, P. S., Pak, S., Rosynek, M. P. & Lunsford, J. H. Reactions of Hydroxyl Radicals on Titania, Silica, Alumina, and Gold Surfaces. *The Journal of Physical Chemistry B* **104**, 2736-2742, doi:10.1021/jp993653e (2000).
- 43 Hewett, K. B., Anderson, L. C., Rosynek, M. P. & Lunsford, J. H. Formation of Hydroxyl Radicals from the Reaction of Water and Oxygen over Basic Metal Oxides. *Journal of the American Chemical Society* **118**, 6992-6997, doi:10.1021/ja960566g (1996).
- 44 Narayanasamy, J. & Kubicki, J. D. Mechanism of Hydroxyl Radical Generation from a Silica Surface: Molecular Orbital Calculations. *The Journal of Physical Chemistry B* **109**, 21796-21807, doi:10.1021/jp0543025 (2005).
- 45 Bogart, K. H. A., Cushing, J. P. & Fisher, E. R. Effects of Plasma Processing Parameters on the Surface Reactivity of OH(X2II) in Tetraethoxysilane/O₂ Plasmas during Deposition of SiO₂. *The Journal of Physical Chemistry B* **101**, 10016-10023, doi:10.1021/jp971596o (1997).
- 46 Sainsbury, T. *et al.* Oxygen Radical Functionalization of Boron Nitride Nanosheets. *Journal of the American Chemical Society* **134**, 18758-18771, doi:10.1021/ja3080665 (2012).
- 47 Belgamwar, R. *et al.* Boron Nitride and Oxide Supported on Dendritic Fibrous Nanosilica for Catalytic Oxidative Dehydrogenation of Propane. *ACS Sustainable Chemistry & Engineering* **8**, 16124-16135, doi:10.1021/acssuschemeng.0c04148 (2020).

The method of calculation is described in the Methods part of the revised manuscript on Page 17:

Microkinetic simulation

The microkinetic simulations of gas-phase conversion were implemented with the Cantera package,⁵³ using the “DTU” model of propane oxidation.³⁷ The technical settings of these simulations were analogous to those used by Kraus and Lindstedt.³¹ Specifically, a constant pressure, ideal gas reactor was used and the time evolution of the gas mixture was modelled using a dynamic time step, adjusted by the solver. All simulations were performed at atmospheric pressure and a temperature of 500°C.

- 31 Kraus, P. & Lindstedt, R. P. It's a Gas: Oxidative Dehydrogenation of Propane over Boron Nitride Catalysts. *J Phys Chem C* **125**, 5623-5634, doi:10.1021/acs.jpcc.1c00165 (2021).
- 37 Hashemi, H., Christensen, J. M., Harding, L. B., Klippenstein, S. J. & Glarborg, P. High-pressure oxidation of propane. *Proceedings of the Combustion Institute* **37**, 461-468, doi:<https://doi.org/10.1016/j.proci.2018.07.009> (2019).
- 53 Cantera: An object-oriented software toolkit for chemical kinetics, thermodynamics, and transport processes v. 2.6.0 (2022).

2. Catalysis studies are not complete; the effect of gas flow, gas ratios, temperature, pressures on product yield and selectivity are not studied.

Reply: Most of these effects were studied and reported in the original manuscript, in particular for SiO₂ as a filler material.

- Temperature variation: Figure 1 and Supplementary Table 2 in the original and in the revised manuscript
- Feed variation: Supplementary Figures 1 and 2 in the original manuscript (1 and 4 in the revised manuscript)
- Contact time variation (gas flows): Supplementary Figure 2 in the original manuscript (Supplementary Figure 4 in the revised version)
- Total flow and bed height: Supplementary Figure 3 in the original manuscript (Supplementary Figure 5 in the revised version)

We added a graph for the determination of the apparent activation energy (Supplementary Figure 2 in the revised version).

Supplementary Fig. 2: Arrhenius plot for the determination of the apparent activation energy for the used materials.

Reaction conditions: T = 470 °C - 510 °C, total flow = 10 ml min⁻¹, feed (C₃H₈/O₂/He) = 30/15/55.

The reaction was also carried out at higher pressure, but under these conditions the lines downstream of the reactor were quickly blocked by polymerization products and we were unsuccessful in making a proper measurement.

The effects observed when varying the reaction parameters are very consistent with those in the literature, as already discussed in the main text of the original manuscript. Thus, we do not gain much new insights here. We would also like to emphasize that a detailed kinetic study was not the aim of the present study. Rather, we wanted to report the formation of PO together with propylene while avoiding CO₂ in a gas phase reaction after ignition of the reaction mixture.

3. Details catalysis stability study for a long duration (say 100 hrs) is needed.

Reply: We have carried out a long-term test with quartz sand as the cheapest filling material. The results were included in the main text on Page 6

The stability of the performance has been proven for SiO₂ for more than 100 h (Supplementary Fig. 3).

and in the Supplementary Information, new Supplementary Figure 3:

Supplementary Fig. 3: Conversion of propane and selectivity to the products (see legend) over silica (α -quartz).

Reaction conditions: T = 500°C, feed (C₃H₈/O₂/He = 30/15/55), total flow rate (10 ml/min), m = 0.666 g.

4. Mechanistic aspects of the catalytic process are weak (only TPD study). DRIFTS or SSNMR or some other spectroscopic studies will provide the required insights.

Reply: To examine the reaction mechanism we performed microkinetic modelling, see our reply to the first point.

Furthermore, the materials were studied before and after catalysis. We decided not to include the data into the Supplementary Information as they do not provide any significantly new aspects compared to the literature and would only unnecessarily expand the manuscript. However, we discuss them in the following to inform the reviewer about the existing results:

Only for *h*-BN were significant differences between the fresh and used catalyst found. Boric acid was detected by XRD in the spent catalyst (Figure R1), but only when the catalyst was used in propane-rich feed over longer times.

Figure R1. XRD of used *h*-BN after propane oxidation; Reaction conditions used (lean): T = 490°C, feed (C₃H₈/O₂/He = 10/5/85), W/F = 0.45 g s ml⁻¹; Reaction conditions used (1st (24 h) and 2nd (120 h) rich): T = 490°C, feed (C₃H₈/O₂/He = 30/15/55), W/F = 0.45 g s ml⁻¹.

The formation of B-OH groups and the deposition of carbonaceous compounds was confirmed by FTIR spectroscopy (Figure R2).

Figure R2. ATR of used *h*-BN after propane oxidation; Reaction conditions used (lean): T = 490°C, feed (C₃H₈/O₂/He = 10/5/85), W/F = 0.45 g s ml⁻¹; Reaction conditions used (1st (24 h) and 2nd (120 h) rich): T = 490°C, feed (C₃H₈/O₂/He = 30/15/55), W/F = 0.45 g s ml⁻¹

Additional confirmation of the oxidation of *h*-BN under the conditions of propane oxidation and the deposition of carbon-containing species was received through photoelectron spectroscopy (Figure R3).

Figure R3. XPS measurement of used *h*-BN (left: C 1s core level; right: O 1s core level) after propane oxidation; Reaction conditions used (lean): T = 490°C, feed (C₃H₈/O₂/He = 10/5/85), W/F = 0.45 g s ml⁻¹; Reaction conditions used (1st (24 h) rich): T = 490°C, feed (C₃H₈/O₂/He = 30/15/55), W/F = 0.45 g s ml⁻¹.

The crystalline surface of *h*-BN appears increasingly amorphous after the reaction (Figure R4).

Figure R4. Fast fourier transform filtered high-resolution transmission electron microscopy of fresh *h*-BN (left) and used *h*-BN (right) after propane oxidation (24 h); Reaction conditions: T = 490°C, feed (C₃H₈/O₂/He = 30/15/55), W/F = 0.45 g s ml⁻¹.

Electron Energy Loss Spectroscopy (EELS) suggests the incorporation of carbon in the surface layers of B (Figure R5).

Figure R5. Local analysis of surface termination of used h-BN after propane oxidation with EELS line scan (left and middle) and corresponding EELS line scan profile (right); Reaction conditions of used h-BN: T = 490°C, feed (C₃H₈/O₂/He = 30/15/55), W/F = 0.45 g s ml⁻¹.

In case of quartz sand, a TPO was performed after long-term testing. Only traces of CO₂ were observed.

We also refer at this point to our response to Point 5 below.

5. Statement “surface-stabilized BO_x species have been thought to be the active sites in a catalyzed reaction at the solid-gas interphase”. This is recently studied in detail here: ACS Sus. Chem. Eng. 2020, 8, 16124.

Reply: In the article mentioned by the reviewer, BN was supported on SiO₂ fibers and found to be partially oxidized to B₂O₃ under the conditions of propane oxidation. The high activity of the catalyst was attributed to the presence of boron oxide. However, no quantitative correlation between boron oxide concentration and activity was established. Therefore, these changes do not necessarily have anything to do with catalysis or the formation of active sites. The formation of boron oxide may just as well be a secondary effect and only indicate the reaction of the feed gas or the products with the filling material. However, an increased concentration of OH-species was found in the most productive catalyst in the cited work. This observation agrees very well with the results of our microkinetic simulation, according to which an increased concentration of OH radicals increases propane conversion. OH radicals and silanol groups can be formed by reaction of water with oxygen in the gas phase, but also by reaction of water with the silica surface (J. Narayanasamy, J. D. Kubicki, The Journal of Physical Chemistry B 2005, 109, 21796-21807. – new Reference 44 in the revised manuscript, please see our answer to your Point 1). The OH radicals can also react with the surface of the filling materials.

We included the following sentence on Page 11 of the revised manuscript and cited the article:

For example, it was found that an increased concentration of OH species on the surface of BN leads to a very productive catalyst.⁴⁷

47 Belgamwar, R. *et al.* Boron Nitride and Oxide Supported on Dendritic Fibrous Nanosilica for Catalytic Oxidative Dehydrogenation of Propane. *ACS Sustainable Chemistry & Engineering* **8**, 16124-16135, doi:10.1021/acssuschemeng.0c04148 (2020).

This MS will have a direct impact on how PO is produced in the industry in the future. However, authors should carry out more detailed studies to get an in-depth insight into this process. Overall, this is an excellent MS and should be accepted for publication after the above-detailed studies.

Reviewer #2 (Remarks to the Author):

The manuscript by Kube et al. reports on the co-production of propylene oxide (PO) and propylene during the thermal oxidation of propane.

Although conceptually this sounds great, in reality, co-production of chemicals can be challenging as the growth rate of both rarely line up. Having said that, the observed chemistry is interesting as – to the best of my knowledge – nobody reported PO formation during these reaction conditions.

Comments:

- In the abstract and later in the manuscript the authors refer to BN and other materials like SiO₂ as being inert. This is not in line with several reports showing that the surface of BN oxidizes under reaction conditions. What the authors probably want to say is that the surface does not play a role in the product evolution (because several materials result in similar product distribution).

Reply: We thank Reviewer 2 for this comment. That is exactly what we want to express. We changed the wording according to the suggestion of Reviewer 2 in the abstract

Surprisingly, we found that the oxidation of propane at elevated temperature over apparently inert materials...

and on Page 6 of the revised manuscript:

These results clearly indicate that the surface of the different materials is not directly catalysing the reaction, but their involvement in initiating or quenching radical reactions cannot be excluded.

- This fact is however at odd with results in Angew. Chem. Int. Ed. 2020, 59(38), 16527-16535. In this paper it is shown that the C₂/C₃ ratio is predominantly controlled by the H-atom that is abstracted in the propane substrate: iso-propyl radicals exclusively form propylene whereas n-propyl can also form ethylene upon the elimination of a methyl radical. The ratio of iso-propyl/n-propyl radicals that one can expect based on radical gas phase chemistry is 1.5/1 whereas the experiments suggest a value closer to 1, based on literature values for the gas phase radical chemistry (also in line with the secondary C-H bonds in propane being weaker than the primary C-H bonds). This suggests that there are additional pathways that lead to this isopropyl-to-n-propyl ratio, possibly on the surface. I do however understand the hypothesis of the authors that the surface would not play a role as several materials gave similar selectivity. I would encourage the authors to reformulate this statement.

Reply: We agree with the reviewer. We showed that also SiO₂, SiC, and an empty reactor show the same reaction performance like *h*-BN with having the same ethylene/propylene ratio trends. However, it is unlikely that on all the different materials identical active sites evolve

under reaction conditions. In particular in an empty reactor the active sites can only be located at the reactor walls.

According to the “NIST Chemical Kinetics Database”, the reaction of propane with oxygen to 1-propyl or iso-propyl radicals, respectively, give the following values:

with A being the pre-exponential factor and n the reaction order.

Based on these values, the iso-propyl/N-propyl ratio is about 1, which is consistent with the experimental value measured in the present work and also indicates that product distribution is mainly based on gas phase reactions.

The Supplementary Figures1 (a and c) and 2 b (4 b in the revised version) show the influence of propane concentration on the selectivity of propylene and ethylene. The figures show that the ratio of ethylene to propylene can be controlled by the inlet concentration of propane, which also speaks for a gas phase mechanism.

To substantiate gas-phase chemistry, we have simulated a microkinetic model of a gas phase reaction. The course of the selectivity is perfectly reproduced by this pure gas phase model. We refer here to our response to Point 1 of Reviewer 1 and the newly inserted section “Microkinetic simulation” on Pages 10-11 of the revised manuscript:

Microkinetic simulation

To obtain deeper insights into the mechanism of propylene oxide formation in these experiments, we turned to microkinetic models of this process. As already noted by Kraus and Lindstedt,³¹ the experimentally observed selectivity towards propylene is fully consistent with a gas-phase mechanism for the conversion. Meanwhile, the predicted selectivity for propylene oxide is somewhat sensitive to the details of the underlying microkinetic model. With the new experiments presented herein, this uncertainty can be clarified, as we find good agreement between experimentally observed selectivity and those predicted for an ideal gas reaction when using the recent “DTU” mechanism for propane oxidation.³⁷ Importantly, this mechanism (unlike the others considered in ref. 31) includes hydroperoxoalkyl chemistry that becomes relevant in the temperature range of this process.²¹

Fig. 3: Microkinetic simulation of the gas-phase reaction.

a, Selectivity (S) with respect to the formation of propylene (C_3H_6) and propylene oxide (c- $\text{C}_3\text{H}_6\text{O}$), versus conversion of propane (X); Shown are results for microkinetic simulations of

an ideal gas reactor at 500°C, using a gas-phase mechanism³⁷ and a feed composition of 30% propane and 15% oxygen; **b**, Conversion of propane (X) as a function of reaction time, for the simulations in **a**; The different traces correspond to the fraction of OH radicals introduced to the feed (solid: 0.0%, dashed: 0.01%, dotted: 0.05%, dash-dotted: 0.1%); The remaining fraction of the feed is composed of N₂.

The good agreement (Fig. 3a) of these ideal gas simulations with the experiments and the fact that experimental selectivity is basically independent of the employed material in the reactor thus clearly point to a gas-phase mechanism of the conversion. This raises the question why different materials nonetheless display different activities. A possible explanation for this is that the materials modify the composition of the gas mixture, *i.e.*, generate radicals that accelerate the gas phase chemistry or quench radicals.^{42,43} To check this hypothesis, we ran additional simulations with modified feed compositions, containing small fractions of [•]OH (Fig. 3b). Here, [•]OH is an exemplary radical species that could plausibly be formed from the couple product water on *h*-BN or SiO₂ surfaces.⁴⁴⁻⁴⁶ For example, it was found that an increased concentration of OH species on the surface of BN leads to a very productive catalyst.⁴⁷ We find that this modification only slightly affects the selectivity at low conversions (Fig. 3a). However, as seen in Fig. 3b, propane conversion is significantly accelerated by the presence of these radicals.

These simulations thus point to a pure gas-phase mechanism for the conversion of propane to propylene and propylene oxide. The filling materials can nonetheless play an important role, by generating radical precursors that can significantly accelerate the conversion. Activities could potentially be further increased by tuning the feed composition and the filling material and by optimizing the process technology.

37 Hashemi, H., Christensen, J. M., Harding, L. B., Klippenstein, S. J. & Glarborg, P. High-pressure oxidation of propane. *Proceedings of the Combustion Institute* **37**, 461-468, doi:<https://doi.org/10.1016/j.proci.2018.07.009> (2019).

42 Suh, M., Bagus, P. S., Pak, S., Rosynek, M. P. & Lunsford, J. H. Reactions of Hydroxyl Radicals on Titania, Silica, Alumina, and Gold Surfaces. *The Journal of Physical Chemistry B* **104**, 2736-2742, doi:10.1021/jp993653e (2000).

43 Hewett, K. B., Anderson, L. C., Rosynek, M. P. & Lunsford, J. H. Formation of Hydroxyl Radicals from the Reaction of Water and Oxygen over Basic Metal Oxides. *Journal of the American Chemical Society* **118**, 6992-6997, doi:10.1021/ja960566g (1996).

44 Narayanasamy, J. & Kubicki, J. D. Mechanism of Hydroxyl Radical Generation from a Silica Surface: Molecular Orbital Calculations. *The Journal of Physical Chemistry B* **109**, 21796-21807, doi:10.1021/jp0543025 (2005).

45 Bogart, K. H. A., Cushing, J. P. & Fisher, E. R. Effects of Plasma Processing Parameters on the Surface Reactivity of OH(X2Π) in Tetraethoxysilane/O₂ Plasmas during Deposition of SiO₂. *The Journal of Physical Chemistry B* **101**, 10016-10023, doi:10.1021/jp971596o (1997).

46 Sainsbury, T. *et al.* Oxygen Radical Functionalization of Boron Nitride Nanosheets. *Journal of the American Chemical Society* **134**, 18758-18771, doi:10.1021/ja3080665 (2012).

47 Belgamwar, R. *et al.* Boron Nitride and Oxide Supported on Dendritic Fibrous Nanosilica for Catalytic Oxidative Dehydrogenation of Propane. *ACS Sustainable Chemistry & Engineering* **8**, 16124-16135, doi:10.1021/acssuschemeng.0c04148 (2020).

The method of calculation is described in the Methods part of the revised manuscript on Page 17:

Microkinetic simulation

The microkinetic simulations of gas-phase conversion were implemented with the Cantera package,⁵³ using the “DTU” model of propane oxidation.³⁷ The technical settings of these simulations were analogous to those used by Kraus and Lindstedt.³¹ Specifically, a constant pressure, ideal gas reactor was used and the time evolution of the gas mixture was modelled using a dynamic time step, adjusted by the solver. All simulations were performed at atmospheric pressure and a temperature of 500°C.

- 31 Kraus, P. & Lindstedt, R. P. It's a Gas: Oxidative Dehydrogenation of Propane over Boron Nitride Catalysts. *J Phys Chem C* **125**, 5623-5634, doi:10.1021/acs.jpcc.1c00165 (2021).
- 37 Hashemi, H., Christensen, J. M., Harding, L. B., Klippenstein, S. J. & Glarborg, P. High-pressure oxidation of propane. *Proceedings of the Combustion Institute* **37**, 461-468, doi:<https://doi.org/10.1016/j.proci.2018.07.009> (2019).
- 53 Cantera: An object-oriented software toolkit for chemical kinetics, thermodynamics, and transport processes v. 2.6.0 (2022).

- Page 5 line 101: the differences in propane consumption rates for the different materials is suggested to be caused by the differences in heat transfer properties of the materials. I do not find the data to judge this statement but it seems unlikely to me that heat transfer would play a role at these small scales and low conversions (something that is also in line with the absence of mass-transfer limitations, as verified by the authors).

Reply: The thermal conductivity values of the materials used were included in the revised Supplementary Information in the Supplementary Table 1.

Supplementary Table 1: Properties of the materials filled into the reactor.

	S_{BET} ($\text{m}^2 \text{g}^{-1}$)	V_p ($\text{cm}^3 \text{g}^{-1}$)	Impurities (wt-%)	Phase composition	Thermal conductivity ($\text{W m}^{-1} \text{K}^{-1}$)
SiO ₂	1.6	0.0004	0.014 Ti 0.003 Mn	α -quartz	7.7-8.4
Aerosil 380	394	1.8	-	Amorphous	0.02
SiC	0.1	0.0006	0.007 Ti 0.002 Mn	mixture of various SiC polytypes main component: moissanite-6H	32-270
h -BN	9.0	0.005	0.7 Si 0.4 Ca 0.2 Zr (0.04) Cr	Hexagonal	220-420

Reference is made to the new Supplementary Table 1 at the appropriate place in the text on Page 5. The high thermal conductivity value for *h*-BN correlates with the high activity of this material. The Aerosil used shows the lowest activity of all tested materials and also has the lowest thermal conductivity. Mass transport and heat transport must be considered

independently of each other here. The heat transfer is the transport of heat from the furnace (heater) through the quartz reactor and the material to the gas. The gas can be heated much more easily if a material with a high thermal conductivity is used in the reactor. The gas heated in this way leads to improved gas phase chemistry. This is also reflected in delayed ignition in the empty reactor (Supplementary Figure 6 in the original Supplementary Information and 8 in the revised version).

- Page 6, line 126: similar observations were reported in *Org. Process Res. Dev.*, 2018, 22 (12), 1644–1652: the observed rate scales linearly with the overall bed volume (more precisely: total reactor contact time), but not with the BN-based contact time. That study also shows how the reaction rate initially increases linear with #BN (keeping the reactor volume constant) and then decreases (volcano-like behavior), indicating radical inhibition (termination) at higher #BN loadings. These observations seem to disprove that the material would not play any role.

Reply: Here we come back to the first point of Reviewer 2. We would like to express that the filler materials are not catalysts in the classical sense. However, this does not mean that they do not influence the prevailing gas phase chemistry.

In the publication mentioned above (<https://pubs.acs.org/doi/abs/10.1021/acs.oprd.8b00301>), the influence of the reactor parameters on the reaction performance was investigated when using *h*-BN as catalyst and SiC as diluent. Our work shows that when SiC is used exclusively, we observe the same reaction performance in propane oxidation (similar selectivity trends) as with *h*-BN, but the propane conversion is smaller (Figure 1 in the original and revised manuscript, orange stars for SiC).

The volcano-like behaviour in Figure 8 of the aforementioned publication was obtained for different *h*-BN/SiC ratios and different flow rates, keeping $W(h\text{-BN})/F(\text{C}_3\text{H}_8)$ constant. The bed volume (total reactor contact time) was kept constant throughout this experiment. Our finding that SiC also has an impact on reaction performance means that it is not justified to use SiC as a diluent.

The observed volcano-like behavior in Figure 8 of the mentioned publication is the result of varying the flow rate while keeping the total bed volume constant. Reducing the flow rate increases the reaction rate to the point where the gas phase chemistry reaches its maximum. Important for gas phase chemistry is the empty space after the material bed, as shown by Kraus *et al.* (<https://pubs.acs.org/doi/pdf/10.1021/acs.jpcc.1c00165>). If the reaction space is shifted to the inside of the material bed by lowering the total flow rate, quenching of the gas phase chemistry by the material itself may occur.

The radical inhibition (termination) observed in the publication at high *h*-BN loadings shows that the material can contribute to the quenching of radical reactions.

- the process simulation is not described in a lot of detail and is not succinct. For instance, page 11, line 199: ...products could be sold for the following hypothetical prices... Does this mean that those numbers were just assumed, or calculated based on a process model (which at this stage would have significant error margins), and if so: what value margin was assumed?

Reply: Thank you for your valuable comment. Only the price for propylene oxide was calculated, while the prices for the other valuable products were taken from the references provided. We agree that this has a significant margin of error at this stage and needs to be optimized to reflect industrial conditions, as the process simulation was performed according to the experimental conditions applied in the laboratory. Nevertheless, we believe that the

process simulation underlines the feasibility of direct propane oxidation even at this stage. We have reworded the process simulation part.

The discussion in the main text on Page 12 of the revised manuscript was changed as follows:

Considering the benefit of the other valuable products produced directly from propane in such a direct process according to their market prices (propylene 1048\$/ton,⁴⁸ hydrogen 6000\$/ton,⁴⁹ and acetaldehyde 973\$/ton⁵⁰) propylene oxide could be sold for the hypothetical price of 3570\$/ton, which is close to the current market price of 2807\$/ton⁵¹. A more detailed description of the process simulation is given in the Supplementary Information.

48 *Price of propylene in 2021*, <<https://www.statista.com/statistics/1170576/price-propylene-forecast-globally/>> (2021).

49 *Deloitte-Ballard Joint White Paper Assesses Hydrogen & Fuel Cell Solutions for Transportation* <<https://www2.deloitte.com/content/dam/Deloitte/cn/Documents/finance/deloitte-cn-fueling-the-future-of-mobility-en-200101.pdf>> (2020).

50 *Price of acetaldehyde from 25.12.2018*, <https://www.echemi.com/productsInformation/pid_Rock16660-acetaldehyde.html> (2018).

51 *Price of propylene oxide from 30.04.2021*, <<http://www.sunsirs.com/uk/prodetail-438.html>> (2021).

In addition, we have added a more detailed description of the process simulation in the Methods part on Pages 18-20 of the revised manuscript and the Supplementary Information (Supplementary Fig. 9, Supplementary Tables 3 and 4 in the revised manuscript) to provide more information about the underlying process:

Supplementary Fig. 9: Aspen HYSYS flow-sheet for direct oxidation of propane to propylene oxide.

Process simulation

The direct oxidation of propane has been simulated using Aspen HYSYS. This process involves the reaction of propane with oxygen to produce propylene oxide as the target product. The rest of the considered reaction products include a mixture of propylene, ethylene, acetaldehyde, hydrogen, water, carbon monoxide and carbon dioxide. The Cubic-Plus-Association (CPA) package has been chosen as the fluid package for the simulation. The CPA property package uses the Cubic-Plus Association equation of state model and is suitable

for the simulation of mixtures containing hydrocarbons, non-hydrocarbons such as carbon dioxide, nitrogen, and polar/associating chemicals such as water, alcohols, glycols, esters or organic acids. The process flow diagram (PFD) corresponding to the direct oxidation of propane simulated in Aspen HYSYS is given in the Supplementary Fig. 9. Three main parts can be distinguished, i.e., conditioning of the feed gases, reaction, and separation of the reaction products and recycling of the unreacted reactants.

The feed gases (propane, oxygen, and helium) are heated up to the reaction temperature (490 °C) by heat exchangers (E-100, E-101, E-102) placed in each of the reactor inlet gas lines. Thereafter, the make-up gas enters the reactor CRV-100, where the reaction between propane and oxygen takes place at 490 °C and 1 bar. Under these conditions the experimental conversion of propane is 40%. The reactor outlet stream consists of unreacted propane and oxygen, helium, propylene oxide, propylene, ethylene, acetaldehyde, hydrogen, water, carbon monoxide and carbon dioxide. Then, different separation stages are carried out to separate the different products and unreacted gases. After cooling the reaction products (E-103), a vapor, liquid and aqueous streams are split in the separator V-100. The vapor stream is mainly composed of ethylene, hydrogen, oxygen, helium, carbon monoxide, and carbon dioxide, whereas most propane, propylene oxide, acetaldehyde, and propylene are recovered in the liquid stream. The vapor stream is further cooled (E-104) and separated into two streams in the separator V-101. The resulting streams are a vapor mixture consisting of helium and hydrogen, and a liquid formed by carbon monoxide, oxygen, and minor amounts of other compounds, such as ethylene, carbon dioxide, propane, and propylene. Helium and hydrogen are separated in V-102 after cooling (E-105). The liquid stream obtained in V-101 is heated (E-106) to recover carbon monoxide and oxygen in the top stream of the separator V-103. After that, oxygen is separated from carbon monoxide in the distillation column T-104 and fed into the reactor together with a fresh oxygen stream. The liquid stream obtained in V-100 is subjected to successive distillation stages to recover propylene, acetaldehyde, propylene oxide and unreacted propane in individual streams. In the distillation column T-100, the light components that could not be separated in V-100 (mainly ethylene, and carbon dioxide) are recovered in the top stream. In the bottom stream, propane, propylene, acetaldehyde, and propylene oxide are obtained. The bottom stream from T-100 enters the distillation column T-101, where propylene is separated in the top stream. The bottom stream from T-101 is fed to the distillation column T-102. Here, the unreacted propane is recovered in the top stream and recirculated to the reactor. Finally, the bottom stream of T-102 is separated into acetaldehyde and propylene oxide in the distillation column T-103.

Supplementary Tab. 3 shows the recovery of the main compounds in their corresponding streams as well as their mole fractions. More than ca. 95 % of propane, propylene oxide, hydrogen, oxygen, propylene, and helium, respectively, are separated and recovered in individual streams. They are high purity streams in which the mole fraction of the corresponding compound is greater than or equal to 0.99, except for the case of the acetaldehyde stream, which has an acetaldehyde mole fraction of ca. 0.95.

Supplementary Tab. 3: Global recovery and mole fraction of the final streams.

Stream in PFD	Global recovery (%) ^a	Mole fraction
Propane	95.73	0.9990
Propylene oxide	99.76	0.9900
Acetaldehyde	96.53	0.9548
Hydrogen	98.63	0.9978
Oxygen	94.59	0.9900
Propylene	94.93	0.9900
Helium	99.97	0.9984

^a Global recovery of comp. *i* (%) = Mole flow of comp. *i* in the stream *i* / Mole flow of comp. *i* in *Product* x 100

The plant cost estimation for the direct oxidation of propane to produce propylene oxide based on the Aspen HYSYS simulation has been performed using the Aspen Process Economic Analyzer (APEA) integrated in Aspen HYSYS. The plant costs can be classified in two major categories, i.e., capital, and operating costs. The capital cost includes the equipment and installed costs. Both costs represent the major fraction of the total capital cost. On the other hand, the costs associated with raw materials and utilities like separation account for the main part of the operating costs. The capital and operating costs have been used to calculate the minimum price at which propylene oxide could be sold, assuming that the capital cost will be completely paid during the first 5 years of the operation of the plant assuming 8000 h of operation per year (Supplementary Tab. 4).

Supplementary Table. 4: Economic analysis of the combined process for propylene and propylene oxide production.

How many years to profit	5	years
Total Capital Cost	17,221,700	\$
Total Operating Cost	13,974,700	\$/year
Total costs in 5 years	87,095,200	\$
Total profit from Hydrogen + Acetaldehyde + Propylene in 5 years	44,234,011	\$
Difference (Investment - Profit)	42,861,189	\$
Total propylene oxide production in 5 years	12,005	ton
Price of propylene oxide from propane dehydrogenation + propene epoxidation	3,570	\$/ton
Commercial price of PO	2,807	\$/ton

The strong point of this manuscript is that the authors observed the formation of PO under these reaction conditions. The authors do however not elaborate on its formation mechanism. I also feel that it is a bit of a stretch to suggest that this approach could replace PO production via traditional pathways. From that perspective I do not think the manuscript is a good fit for Nature Communications.

Reply: To support our proposal for a gas phase mechanism, we have performed a simulation using a microkinetic model for a gas phase mechanism, which strongly supports our assumption (see above and our reply Point 1 of Reviewer 1).

It is usually a long way from the first experimental finding in the laboratory to the introduction of a new industrial process. But that is no reason not to publish this finding and

the idea of a direct process. Furthermore, based on our process engineering simulation, we are convinced that the process has potential. This is also confirmed by the interest in the pre-publication of our manuscript on ChemRxiv. Cambridge: Cambridge Open Engage; 2022; DOI: 10.26434/chemrxiv-2022-2clnw.

Reviewer #3 (Remarks to the Author):

Green synthesis of propylene oxide directly from propane
2-17 refers to page 2, line 17. Etc

I. General comments

This is an interesting report that would be a valuable addition to the literature. There are a few comments and requests for clarification below.

II. Main comments / questions

a. Fig 1: there appears to be some variation in selectivity performance, specifically in c & d. Is this due to the large spread in temperatures that were used (470 to 510 oC)? Perhaps the data could be further clarified by adding figures in the SI

Reply: The variations in selectivity in Figure 1 c are due to the measurement accuracy of the detectors used. Propylene and propylene oxide were analyzed with the highest accuracy using a Flame Ionization Detector (FID). The products in Figure 1c were partially analyzed (open symbols, CO) and in Figure 1d completely analyzed with a Thermal Conductivity Detector (TCD). The FID shows higher sensitivity than the TCD.

b. Fig 1: why is BN the only material studied at higher conversions? Is it possible to get the same results by increasing residence time over the other materials?

Reply: The data is difficult to measure because the materials are not catalysts in the classical sense. Thus, the conversion cannot be directly increased by increasing the mass or decreasing the flow. Nevertheless, the material has an influence on the propane conversion for the reasons discussed above in the responses to Reviewers 1 and 2. For example, we have added the following remark on Page 6 of the revised manuscript:

These results clearly indicate that the surface of the different materials is not directly catalysing the reaction, but their involvement in initiating or quenching radical reactions cannot be excluded.

c. Were blank tests (with only quartz tubes, no silica wool) run?

Reply: We performed an experiment with the completely empty tube (Supplementary Figure 6 in the original Supplementary Information and Supplementary Figure 8 in the revised version).

d. 98: PO forms only when there is incomplete consumption of O₂? Have experiments been performed on higher initial O₂ concentrations and selectivity evaluated?

Reply: Experiments with higher initial O₂ concentrations were not carried out because of the explosive character of such reaction mixtures. Such experiments must be carried out at least in a two-stage reactor. After almost all the oxygen has been consumed in the first stage, additional oxygen could be added to the effluent gas before the reaction mixture enters the second stage.

e. 105: Are the BOx active sites why higher conversions are reported for hBN only?

Reply: Thank you very much, the question is very valid. As already stated above in the answers to the other reviewers, we are always on the same S-X-plot with all materials, but the propane conversion is different. Since we have evidence of a gas phase mechanism based on our microkinetic simulation, we attribute the differences to differences in the thermal conductivity of the materials and now report these in the revised Supplementary Table 1 (see our Reply to Reviewer 2 on Pages 12-13 of this document).

Supplementary Table 1: Properties of the materials filled into the reactor.

	S_{BET} ($\text{m}^2 \text{g}^{-1}$)	V_p ($\text{cm}^3 \text{g}^{-1}$)	Impurities (wt-%)	Phase composition	Thermal conductivity ($\text{W m}^{-1} \text{K}^{-1}$)
SiO ₂	1.6	0.0004	0.014 Ti 0.003 Mn	α -quartz	7.7-8.4
Aerosil 380	394	1.8	-	Amorphous	0.02
SiC	0.1	0.0006	0.007 Ti 0.002 Mn	mixture of various SiC polytypes main component: moissanite-6H	32-270
h -BN	9.0	0.005	0.7 Si 0.4 Ca 0.2 Zr (0.04) Cr	Hexagonal	220-420

With higher thermal conductivity of the filling material, the heat from the furnace is better transferred to the reacting molecules. A clear indication of this is the delayed ignition behavior at 490°C in the empty reactor (Supplementary Fig. 6 in the original Supplementary Information, Supplementary Fig. 8 in the revised version). Furthermore, we do not exclude that the radical reactions in the gas phase are mediated by the surface and that here the different surfaces have different properties in the generation of radicals (see new Figure 3b in the revised manuscript) or in the quenching of radical reactions. See the chapter on microkinetic simulation on Pages 10 and 11 of the revised manuscript.

Microkinetic simulation

To obtain deeper insights into the mechanism of propylene oxide formation in these experiments, we turned to microkinetic models of this process. As already noted by Kraus and Lindstedt,³¹ the experimentally observed selectivity towards propylene is fully consistent with a gas-phase mechanism for the conversion. Meanwhile, the predicted selectivity for propylene oxide is somewhat sensitive to the details of the underlying microkinetic model. With the new experiments presented herein, this uncertainty can be clarified, as we find good agreement between experimentally observed selectivity and those predicted for an ideal gas reaction when using the recent “DTU” mechanism for propane oxidation.³⁷ Importantly, this mechanism (unlike the others considered in ref. 31) includes hydroperoxoalkyl chemistry that becomes relevant in the temperature range of this process.²¹

Fig. 3: Microkinetic simulation of the gas-phase reaction.

a, Selectivity (S) with respect to the formation of propylene (C_3H_6) and propylene oxide ($c-C_3H_6O$), versus conversion of propane (X); Shown are results for microkinetic simulations of an ideal gas reactor at $500^\circ C$, using a gas-phase mechanism³⁷ and a feed composition of 30% propane and 15% oxygen; **b**, Conversion of propane (X) as a function of reaction time, for the simulations in **a**; The different traces correspond to the fraction of OH radicals introduced to the feed (solid: 0.0%, dashed: 0.01%, dotted: 0.05%, dash-dotted: 0.1%); The remaining fraction of the feed is composed of N_2 .

The good agreement (Fig. 3a) of these ideal gas simulations with the experiments and the fact that experimental selectivity is basically independent of the employed material in the reactor thus clearly point to a gas-phase mechanism of the conversion. This raises the question why different materials nonetheless display different activities. A possible explanation for this is that the materials modify the composition of the gas mixture, *i.e.*, generate radicals that accelerate the gas phase chemistry or quench radicals.^{42,43} To check this hypothesis, we ran additional simulations with modified feed compositions, containing small fractions of $\cdot OH$ (Fig. 3b). Here, $\cdot OH$ is an exemplary radical species that could plausibly be formed from the couple product water on $h-BN$ or SiO_2 surfaces.⁴⁴⁻⁴⁶ For example, it was found that an increased concentration of OH species on the surface of BN leads to a very productive catalyst.⁴⁷ We find that this modification only slightly affects the selectivity at low conversions (Fig. 3a). However, as seen in Fig. 3b, propane conversion is significantly accelerated by the presence of these radicals.

These simulations thus point to a pure gas-phase mechanism for the conversion of propane to propylene and propylene oxide. The filling materials can nonetheless play an important role, by generating radical precursors that can significantly accelerate the conversion. Activities could potentially be further increased by tuning the feed composition and the filling material and by optimizing the process technology.

37 Hashemi, H., Christensen, J. M., Harding, L. B., Klippenstein, S. J. & Glarborg, P. High-pressure oxidation of propane. *Proceedings of the Combustion Institute* **37**, 461-468, doi:<https://doi.org/10.1016/j.proci.2018.07.009> (2019).

42 Suh, M., Bagus, P. S., Pak, S., Rosynek, M. P. & Lunsford, J. H. Reactions of Hydroxyl Radicals on Titania, Silica, Alumina, and Gold Surfaces. *The Journal of Physical Chemistry B* **104**, 2736-2742, doi:10.1021/jp993653e (2000).

43 Hewett, K. B., Anderson, L. C., Rosynek, M. P. & Lunsford, J. H. Formation of Hydroxyl Radicals from the Reaction of Water and Oxygen over Basic Metal Oxides.

- Journal of the American Chemical Society* **118**, 6992-6997, doi:10.1021/ja960566g (1996).
- 44 Narayanasamy, J. & Kubicki, J. D. Mechanism of Hydroxyl Radical Generation from a Silica Surface: Molecular Orbital Calculations. *The Journal of Physical Chemistry B* **109**, 21796-21807, doi:10.1021/jp0543025 (2005).
- 45 Bogart, K. H. A., Cushing, J. P. & Fisher, E. R. Effects of Plasma Processing Parameters on the Surface Reactivity of OH(X2Π) in Tetraethoxysilane/O₂ Plasmas during Deposition of SiO₂. *The Journal of Physical Chemistry B* **101**, 10016-10023, doi:10.1021/jp971596o (1997).
- 46 Sainsbury, T. *et al.* Oxygen Radical Functionalization of Boron Nitride Nanosheets. *Journal of the American Chemical Society* **134**, 18758-18771, doi:10.1021/ja3080665 (2012).
- 47 Belgamwar, R. *et al.* Boron Nitride and Oxide Supported on Dendritic Fibrous Nanosilica for Catalytic Oxidative Dehydrogenation of Propane. *ACS Sustainable Chemistry & Engineering* **8**, 16124-16135, doi:10.1021/acssuschemeng.0c04148 (2020).

The method of calculation is described in the Methods part of the revised manuscript on Page 17:

Microkinetic simulation

The microkinetic simulations of gas-phase conversion were implemented with the Cantera package,⁵³ using the “DTU” model of propane oxidation.³⁷ The technical settings of these simulations were analogous to those used by Kraus and Lindstedt.³¹ Specifically, a constant pressure, ideal gas reactor was used and the time evolution of the gas mixture was modelled using a dynamic time step, adjusted by the solver. All simulations were performed at atmospheric pressure and a temperature of 500°C.

- 31 Kraus, P. & Lindstedt, R. P. It's a Gas: Oxidative Dehydrogenation of Propane over Boron Nitride Catalysts. *J Phys Chem C* **125**, 5623-5634, doi:10.1021/acs.jpcc.1c00165 (2021).
- 37 Hashemi, H., Christensen, J. M., Harding, L. B., Klippenstein, S. J. & Glarborg, P. High-pressure oxidation of propane. *Proceedings of the Combustion Institute* **37**, 461-468, doi:<https://doi.org/10.1016/j.proci.2018.07.009> (2019).
- 53 Cantera: An object-oriented software toolkit for chemical kinetics, thermodynamics, and transport processes v. 2.6.0 (2022).

f. 111: Are similar activation energies expected for the different stable intermediates? (in relation to 211) or were they higher (117)?

Reply: Based on the available data, it is not possible to determine the activation energies of the intermediates. A detailed kinetic study was also not the aim of the work presented here.

g. 214: The sentence starting with “redox active catalysts” is difficult to interpret. It would seem that both approaches are rather inappropriate for the formation of PO?

Reply: Propylene oxide is a sensitive product that undergoes rapid further oxidation. We think that it is easier to generate propylene oxide in propane-rich feed composition and at a temperature of 490°C in the gas phase than to optimize a redox-active catalyst so that it no longer catalyses the subsequent reactions of propylene oxide on the surface to CO₂. We have

reworded the relevant section on Page 13 of the revised manuscript and hope that our statement is now better understood:

We think that it is easier to generate propylene oxide in propane-rich feed composition and at a temperature of 490°C in the gas phase than to optimize a redox-active catalyst and the process conditions so that subsequent reactions of propylene oxide on the surface does not occur.

h. 218: how can you prove that a selective catalytic surface will not give rise to additional levels of success? It is well known that controlling the residence time distribution in partial oxidation catalysis (to remove products quickly from the reactor without overoxidation) gives rise to better yields. It is possible to combine the concepts of selective surface reaction (high specificity for target molecule, low specificity for product molecule) along with short residence times.

Reply: Yes, of course. That is a good point. Literature on such approaches was already cited in the original manuscript, such as the work of Lanny Schmidt (references 38-41 in the revised manuscript). However, the solution proposed in the present work is very simple, as catalyst development is largely unnecessary. But this does not exclude the possibility of further optimization of catalyst properties and process conditions in order to achieve even better results. We agree with Reviewer 3 and inserted the following sentence on Page 11 of the revised manuscript:

These simulations thus point to a pure gas-phase mechanism for the conversion of propane to propylene and propylene oxide. The filling materials can nonetheless play an important role, by generating radical precursors that can significantly accelerate the conversion. Activities could potentially be further increased by tuning the feed composition and the filling material.

i. Have other oxidative PDH researchers (for example, those studying BN) reported higher yields of propylene than reported here? If not, that would give further credit to the arguments presented here. Many of these prior research groups were likely making PO as a product but did not have the proper GC systems to detect & analyze?

Reply: No, the propylene yields we observed are similar to other researchers working on oxidative PDH with BN- or boron-based materials. This suggests that in these works the mass balances have not been checked or that other reasons, such as polymerisation or post-combustion on hot parts of the apparatus after the reactor have led to the loss of propylene oxide.

j. Regarding the techno economic analysis, I do not think that the authors accounted for any separation costs. How could one separate the product mixture that contains O₂ in addition to hydrocarbons?

Reply: Thank you for your comment. Separation costs are included in the operating costs in addition to the costs for raw materials. In the mentioned case cooling and subsequent separation via a flash was considered. To enable the reader to follow the process simulation we added a more detailed description in the Methods part and the Supplementary Information (Supplementary Fig. 9 and Supplementary Tables 3 and 4) to provide further information on the procedure it is based on. Please see also our reply the Reviewer 2 on Pages 13-16 of this document.

Supplementary Fig. 9: Aspen HYSYS flow-sheet for direct oxidation of propane to propylene oxide.

Process simulation

The direct oxidation of propane has been simulated using Aspen HYSYS. This process involves the reaction of propane with oxygen to produce propylene oxide as the target product. The rest of the considered reaction products include a mixture of propylene, ethylene, acetaldehyde, hydrogen, water, carbon monoxide and carbon dioxide. The Cubic-Plus-Association (CPA) package has been chosen as the fluid package for the simulation. The CPA property package uses the Cubic-Plus Association equation of state model and is suitable for the simulation of mixtures containing hydrocarbons, non-hydrocarbons such as carbon dioxide, nitrogen, and polar/associating chemicals such as water, alcohols, glycols, esters or organic acids. The process flow diagram (PFD) corresponding to the direct oxidation of propane simulated in Aspen HYSYS is given in the Supplementary Fig. 9. Three main parts can be distinguished, i.e., conditioning of the feed gases, reaction, and separation of the reaction products and recycling of the unreacted reactants.

The feed gases (propane, oxygen, and helium) are heated up to the reaction temperature (490 °C) by heat exchangers (E-100, E-101, E-102) placed in each of the reactor inlet gas lines. Thereafter, the make-up gas enters the reactor CRV-100, where the reaction between propane and oxygen takes place at 490 °C and 1 bar. Under these conditions the experimental conversion of propane is 40%. The reactor outlet stream consists of unreacted propane and oxygen, helium, propylene oxide, propylene, ethylene, acetaldehyde, hydrogen, water, carbon monoxide and carbon dioxide. Then, different separation stages are carried out to separate the different products and unreacted gases. After cooling the reaction products (E-103), a vapor, liquid and aqueous streams are split in the separator V-100. The vapor stream is mainly composed of ethylene, hydrogen, oxygen, helium, carbon monoxide, and carbon dioxide, whereas most propane, propylene oxide, acetaldehyde, and propylene are recovered in the liquid stream. The vapor stream is further cooled (E-104) and separated into two streams in the separator V-101. The resulting streams are a vapor mixture consisting of helium and hydrogen, and a liquid formed by carbon monoxide, oxygen, and minor amounts of other compounds, such as ethylene, carbon dioxide, propane, and propylene. Helium and hydrogen are separated in V-102 after cooling (E-105). The liquid stream obtained in V-101 is heated (E-106) to recover carbon monoxide and oxygen in the top stream of the separator V-103. After that, oxygen is separated from carbon monoxide in the distillation column T-104 and fed into the reactor together with a fresh oxygen stream. The liquid stream obtained in V-100 is subjected to successive distillation stages to recover propylene, acetaldehyde, propylene oxide and unreacted propane in individual streams. In the distillation column T-100, the light

components that could not be separated in V-100 (mainly ethylene, and carbon dioxide) are recovered in the top stream. In the bottom stream, propane, propylene, acetaldehyde, and propylene oxide are obtained. The bottom stream from T-100 enters the distillation column T-101, where propylene is separated in the top stream. The bottom stream from T-101 is fed to the distillation column T-102. Here, the unreacted propane is recovered in the top stream and recirculated to the reactor. Finally, the bottom stream of T-102 is separated into acetaldehyde and propylene oxide in the distillation column T-103.

Supplementary Tab. 3 shows the recovery of the main compounds in their corresponding streams as well as their mole fractions. More than ca. 95 % of propane, propylene oxide, hydrogen, oxygen, propylene, and helium, respectively, are separated and recovered in individual streams. They are high purity streams in which the mole fraction of the corresponding compound is greater than or equal to 0.99, except for the case of the acetaldehyde stream, which has an acetaldehyde mole fraction of ca. 0.95.

Supplementary Tab. 3: Global recovery and mole fraction of the final streams.

Stream in PFD	Global recovery (%) ^a	Mole fraction
Propane	95.73	0.9990
Propylene oxide	99.76	0.9900
Acetaldehyde	96.53	0.9548
Hydrogen	98.63	0.9978
Oxygen	94.59	0.9900
Propylene	94.93	0.9900
Helium	99.97	0.9984

^a Global recovery of comp. *i* (%) = Mole flow of comp. *i* in the stream *i* / Mole flow of comp. *i* in *Product* x 100

The plant cost estimation for the direct oxidation of propane to produce propylene oxide based on the Aspen HYSYS simulation has been performed using the Aspen Process Economic Analyzer (APEA) integrated in Aspen HYSYS. The plant costs can be classified in two major categories, i.e., capital, and operating costs. The capital cost includes the equipment and installed costs. Both costs represent the major fraction of the total capital cost. On the other hand, the costs associated with raw materials and utilities like separation account for the main part of the operating costs. The capital and operating costs have been used to calculate the minimum price at which propylene oxide could be sold, assuming that the capital cost will be completely paid during the first 5 years of the operation of the plant assuming 8000 h of operation per year (Supplementary Tab. 4).

Supplementary Table. 4: Economic analysis of the combined process for propylene and propylene oxide production.

How many years to profit	5	years
Total Capital Cost	17,221,700	\$
Total Operating Cost	13,974,700	\$/year
Total costs in 5 years	87,095,200	\$
Total profit from Hydrogen + Acetaldehyde + Propylene in 5 years	44,234,011	\$
Difference (Investment - Profit)	42,861,189	\$
Total propylene oxide production in 5 years	12,005	ton
Price of propylene oxide from propane dehydrogenation + propene epoxidation	3,570	\$/ton
Commercial price of PO	2,807	\$/ton

III. Stylistic comments or typos

a. 117: small letter “addition”

Reply: Thank you, corrected.

REVIEWERS' COMMENTS

Reviewer #1 (Remarks to the Author):

MS is thoroughly revised and my as well as other reviewers' comments were answered in-depth and supported by new experiments and data.
I recommend its acceptance in NatureComm.

Reviewer #2 (Remarks to the Author):

In this manuscript the authors demonstrate the potential to co-produce propylene and propylene oxide via the oxidation of propane in the gas-phase at intermediate temperatures. They show identical selectivity vs conversion plots for a variety of fillers as well as an empty reactor, highlighting the limited role of the surface in propagating the reactions. Instead, they show through microkinetic simulation that the role of the surface in elevating conversion is likely due to the generation of radicals that contribute to accelerating gas-phase reactions. Finally, they perform preliminary techno-economic comparison to highlight the direct epoxidation from propane to propylene as a potentially feasible route, comparable to current technology.

Strong Points:

- Clear selectivity vs. conversion plots with a variety of materials to highlight underlying point
- Reasonable use of microkinetic model to demonstrate the plausible influence of gas-phase initiators as well as PO production from a purely gas-phase mechanism in line with observed selectivity distribution
- Use of techno-economic analysis to highlight the significance of the finding

Critique:

- The authors could do more to consider the role of the filler in influencing propane conversion and selectivity. Why is hBN, etc. capable of initiating and quenching radicals do not over-oxidizing PO?

Reviewer #3 (Remarks to the Author):

I am happy with the corrections

Reply to Reviewers' comments

We thank the reviewers for reviewing our manuscript again and for the positive feedback.

Reviewer #2 (Remarks to the Author):

In this manuscript the authors demonstrate the potential to co-produce propylene and propylene oxide via the oxidation of propane in the gas-phase at intermediate temperatures. They show identical selectivity vs conversion plots for a variety of fillers as well as an empty reactor, highlighting the limited role of the surface in propagating the reactions. Instead, they show through microkinetic simulation that the role of the surface in elevating conversion is likely due to the generation of radicals that contribute to accelerating gas-phase reactions. Finally, they perform preliminary technoeconomic comparison to highlight the direct epoxidation from propane to propylene as a potentially feasible route, comparable to current technology.

Strong Points:

- Clear selectivity vs. conversion plots with a variety of materials to highlight underlying point
- Reasonable use of microkinetic model to demonstrate the plausible influence of gas-phase initiators as well as PO production from a purely gas-phase mechanism in line with observed selectivity distribution
- Use of technoeconomic analysis to highlight the significance of the finding

Critique:

- The authors could do more to consider the role of the filler in influencing propane conversion and selectivity. Why is hBN, etc. capable of initiating and quenching radicals do not over-oxidizing PO?

Reply: Our microkinetic simulations indicate a pure gas-phase mechanism for the conversion of propane to propylene and propylene oxide. The radicals are therefore part of the mechanism anyway and do not have a negative effect on the selectivity. It's different if you have redox active centers, like vanadium oxide, in the catalyst, then overoxidation can occur due to surface reactions.